# *Mycobacterium tuberculosis* strain with deletions in *menT3* and *menT4* is attenuated and confers protection in mice and guinea pigs

Tannu Priya Gosain[1], Saurabh Chugh [1], Zaigham Abbas Rizvi[2], Neeraj Kumar Chauhan[1], Saqib Kidwai[1], Krishan Gopal Thakur [3], Amit Awasthi[2] & Ramandeep Singh [1] ✉

The genome of *Mycobacterium tuberculosis* encodes for a large repertoire of toxin-antitoxin systems. In the present study, MenT3 and MenT4 toxins belonging to MenAT subfamily of TA systems have been functionally characterized. We demonstrate that ectopic expression of these toxins inhibits bacterial growth and this is rescued upon co-expression of their cognate antitoxins. Here, we show that simultaneous deletion of *menT3* and *menT4* results in enhanced susceptibility of *M. tuberculosis* upon exposure to oxidative stress and attenuated growth in guinea pigs and mice. We observed reduced expression of transcripts encoding for proteins that are essential or required for intracellular growth in mid-log phase cultures of Δ*menT4ΔT3* compared to parental strain. Further, the transcript levels of proteins involved in efficient bacterial clearance were increased in lung tissues of Δ*menT4ΔT3* infected mice relative to parental strain infected mice. We show that immunization of mice and guinea pigs with Δ*menT4ΔT3* confers significant protection against *M. tuberculosis* infection. Remarkably, immunization of mice with Δ*menT4ΔT3* results in increased antigen-specific $T_H1$ bias and activated memory T cell response. We conclude that MenT3 and MenT4 are important for *M. tuberculosis* pathogenicity and strains lacking *menT3* and *menT4* have the potential to be explored further as vaccine candidates.

TA systems are small genetic elements that are prevalent in most prokaryote genomes[1–3]. TA systems comprise two genes that encode for a stable toxin and unstable antitoxin[1–7]. TA systems have been classified into eight types based on the nature of antitoxin (protein or RNA) and the mechanisms by which toxin activity is neutralized[2,8]. In type I, III and VIII TA systems, antitoxins are small non-coding RNAs while in the remaining subfamilies, antitoxins are proteins[2,8]. The toxins belonging to type I–type VII TA systems are proteinaceous in nature. However, in the case of the type VIII TA system, the toxin is a small RNA[2,8]. The expression of toxins belonging to TA systems inhibits bacterial growth in either bactericidal or bacteriostatic manner by targeting essential cellular processes such as transcription, replication,

[1]Centre for Tuberculosis Research, Translational Health Sciences and Technology Institute, NCR Biotech Science Cluster, 3rd Milestone, Faridabad-Gurugram Expressway, Faridabad 121001, India. [2]Centre for Immunobiology and Immunotherapy, Translational Health Sciences and Technology Institute, NCR Biotech Science Cluster, 3rd Milestone, Faridabad-Gurugram Expressway, Faridabad 121001, India. [3]Structural Biology Laboratory, Council of Scientific and Industrial Research-Institute of Microbial Technology (CSIR-IMTECH), Chandigarh 160036, India. ✉e-mail: ramandeep@thsti.res.in

translation, cell wall biosynthesis, membrane integrity and cytoskeleton formation[1,9]. These systems have been demonstrated to contribute to plasmid maintenance, abortive phage infection, antibiotic persistence and pathogenesis[1,10]. The detailed phylogenetic and bioinformatic analysis revealed that *M. tuberculosis* genome encodes for ≥ 90 TA systems, which are highly conserved in members of the *M. tuberculosis* complex[11–14]. Most of these belong to various subfamilies of type II TA systems such as VapBC, MazEF, RelBE, HigBA, ParDE, HicAB, MbcAT, PezAT, DarTG and Rse-Xre. The toxins belonging to type II TA systems inhibit *M. tuberculosis* growth by cleaving either mRNA or tRNA or rRNA or degrading NAD⁺ or inhibiting DNA gyrase activity or by ADP ribosylation of single-stranded DNA[15–20]. Additionally, subsets of TA systems exhibit differential expression patterns upon *M. tuberculosis* exposure to stress conditions[12,21,22]. Many of these systems are dispensable for the survival of *M. tuberculosis* in stress conditions, thereby indicating that these modules might function cumulatively and contribute to stress adaptation[18,21,23–25]. We have previously reported that TA systems or toxins belonging to the type II subfamily are essential for *M. tuberculosis* to establish infection in mice or guinea pigs[21,23,25–27].

The genome of *M. tuberculosis* encodes for four proteins belonging to the DUF1814 family of nucleotidyl transferases[11,28]. These proteins, MenT1, MenT2, MenT3 and MenT4, share sequence homology with toxins from type IV TA systems. The antitoxins and toxins of type IV TA systems do not interact but compete for binding to the cellular target[1,2,8]. MenT proteins harbor four highly conserved motifs, including nucleotidyl transferase (NTase) like domain. Motifs I and II are present at the amino-terminus and comprise of hG[G/S]x₉₋₁₃DhD domain. Due to its similarity to RxxRxxR observed in tRNA NTases, motif III, KLxAaxxR is predicted to be involved in base stacking interactions for incoming nucleotides. Motif IV comprises a pentad of conserved amino acids +DxxD. Studies have shown that mutation of highly conserved residues in MenT3 (G62 in motif I, D82 in motif II, K189 in motif III and D208 in motif IV) abolishes its growth inhibition activity in *E. coli*[29]. The three-dimensional structures of MenT1, MenT3 and MenT4 toxins from *M. tuberculosis* have been solved at a resolution of 1.65 Å, 1.6 Å and 1.2 Å, respectively[30,31]. These toxins feature a common toxin fold and are bilobed globular proteins[30]. A more detailed analysis revealed that the overall architectures of MenT3 and MenT4 are similar with a root mean square deviation (RMSD) of 4.7 Å. Upon superimposition of the two structures, the authors observed that the active site residues of MenT3 (D80, K189 and D211) and MenT4 (D69, K171 and D186) were at a similar position[30]. Superimposition of MenT1 with MenT3 or MenT4 resulted in an RMSD of 3.829 Å and 4.232 Å, respectively, and a similar alignment of core regions[31]. In another study it has been shown that MenA3 phosphorylates and inactivates the cognate toxin, MenT3[29]. It has also been reported that MenT1, MenT3 and MenT4 possess nucleotidyl transferase activity[30,31]. MenT3 displays a preference for pyrimidines and modifies *M. tuberculosis* tRNA^Ser isoacceptors[30]. Cai et al. demonstrated that MenT3 also weakly modifies tRNA^Leu. Recently, it has been reported that MenT4 exhibits a preference for GTP and modifies several tRNAs, including tRNA^Ser[31]. Taken together, these findings suggest that overexpression of MenT toxins results in growth inhibition by preventing aminoacylation and tRNA charging[30,31].

Despite significant advancements in the characterization of type II TA systems, very limited information is available about the role of MenAT TA systems in the physiology and pathogenesis of *M. tuberculosis*. In the present study, we have performed experiments to investigate the contribution of MenT3 and MenT4 in *M. tuberculosis* physiology and pathogenesis. We show that MenT3 and MenT4 are mutually redundant and simultaneous deletion of *menT3* and *menT4* results in increased susceptibility to oxidative stress and severe attenuation of *M. tuberculosis* in mice and guinea pigs. Using host RNA-seq, we demonstrate that transcripts encoding for proteins involved in calcium signaling, immune responses, apoptosis, and autophagy were differentially expressed in lung tissues of mice infected with Δ*menT4*Δ*T3* relative to parental strain infected mice. We also demonstrate that immunization with Δ*menT4*Δ*T3* strain (1) imparts significant protection against *M. tuberculosis* in mice and guinea pigs and (2) increases antigen-specific T_H1 immune responses and activated memory T-cell response in mice compared to naive mice. Taken together, this study has enhanced our understanding of the contribution of toxins belonging to MenAT TA systems in mycobacterial physiology and pathogenesis.

## Results

### Ectopic expression of MenT3 and MenT4 results in growth inhibition

In order to investigate the effect of overexpression of MenT3 or MenT4 on the growth of *E. coli* or *M. tuberculosis*, wild type toxins or their mutants were individually cloned in either pET28b (IPTG inducible) or pTetR (Atc inducible) expression vectors. As reported earlier, overexpression of MenT3 or MenT4 inhibited *E. coli* growth in comparison to uninduced cultures (Fig. 1a, b)[30]. Further, G62, K189, D208 in motifs I, III and IV of MenT3 and G51, D71 in motifs I and II of MenT4 were mutated, cloned in pET28b and growth assays were performed (Fig. S1a). As expected, overexpression of either MenT3^G62A or MenT3^K189A or MenT3^D208A or MenT4^G51A or MenT4^D71A proteins did not inhibit *E. coli* growth (Figs. 1a, b and S1b). Next, to verify whether the co-expression of antitoxins can alleviate the growth inhibition activity of MenT3 and MenT4 toxins, we cloned them along with their cognate or non-cognate antitoxin in MCS-I and MCS-II of an IPTG inducible expression system, pETDuet. We observed that growth inhibition associated with MenT3 and MenT4 overexpression was restored by co-expression of their cognate antitoxins, MenA3 and MenA4, respectively (Fig. 1c, d). However, no growth restoration was seen upon co-expression of MenT3 and MenT4 with their non-cognate antitoxins (Fig. 1c, d). In agreement with *E. coli* data, we observed that inducible expression of either MenT3 or MenT4 also inhibited *M. tuberculosis* growth (Fig. 1e). In comparison to strain harboring vector control, overexpression of MenT3 and MenT4 reduced the bacterial growth by ~ 5.0-fold at 4 days post-induction (Fig. 1f). This growth defect upon overexpression of either MenT3 or MenT4 increased to ~28.0- and 15.0-fold, respectively, at 7 days post-induction (Fig. 1f).

### Deletion of both *menT3* and *menT4* increases the susceptibility of *M. tuberculosis* to oxidative stress

The abundance of TA systems in members of the *M. tuberculosis* complex raises a possibility that these might function in a cumulative manner and contribute to *M. tuberculosis* survival in stress conditions and host tissues[11,12,14]. In order to understand the role of MenT3 and MenT4 in the physiology of *M. tuberculosis*, we constructed Δ*menT3*, Δ*menT4* single mutant and Δ*menT4*Δ*T3* double mutant strain using temperature-sensitive mycobacteriophages (Fig. 1g). As shown in Fig. 1h, PCR amplification using *menT3* locus-specific primers resulted in amplifications of sizes 882 bp, 1.5 kb and 1.5 kb using genomic DNA isolated from wild type, Δ*menT3* and Δ*menT4*Δ*T3* strains of *M. tuberculosis*, respectively. PCR amplification of sizes 885 bp, 2.0 kb and 2.0 kb were obtained from genomic DNA isolated from wild type, Δ*menT4* and Δ*menT4*Δ*T3* strains of *M. tuberculosis*, respectively, using *menT4* locus primers (Fig. 1h). Whole genome sequencing revealed that sequences aligning to *menT3* and *menT4* locus were missing from the reads obtained from Δ*menT4*Δ*T3* genomic DNA in comparison to reads obtained from the wild type strain (Fig. S2). These observations confirmed that the open reading frame for *menT3* and *menT4* has been replaced with kanamycin and hygromycin resistance gene, respectively, in the Δ*menT4*Δ*T3* strain of *M. tuberculosis*.

We next compared the growth patterns of wild type and Δ*menT4*Δ*T3 M. tuberculosis* strains in liquid culture. We observed that both strains displayed comparable bacterial counts during various

stages of growth in vitro. We also compared the survival of parental and Δ*menT4*Δ*T3* strains after exposure to various stress conditions. As shown in Fig. 2a, relative to the parental strain, Δ*menT4*Δ*T3* strain was ~8.5-fold more susceptible to oxidative stress after exposure for 24 h. However, at 72 h post-exposure to oxidative stress, Δ*menT4*Δ*T3* showed ~ 225.0-fold increased susceptibility in comparison to the parental strain (Fig. 2a). We also observed that complementation of the double mutant strain with *menT3* partially restored this growth defect (Fig. S3a). However, we were unable to restore this growth defect of double mutant strain upon complementation with *menT4* (Fig. S3a). qPCR studies revealed that the transcript levels for *menT3* and *menT4* were restored in their respective single-complemented strain (Fig. S5a). We also found that the sensitivity of wild type, Δ*menT3* and Δ*menT4* was comparable after being exposed to oxidative stress (Fig. S3c, d). Previously, it has been demonstrated that a subset of toxins is induced in response to stress conditions such as low oxygen, nutrient limiting, macrophage engulfment or drug exposure. The increased transcription and synthesis of toxins lead to TA systems activation and subsequent growth inhibition[12,21,22,32]. Since Δ*menT4*Δ*T3* was susceptible upon

exposure to oxidative stress relative to the parental strain, we next measured the relative levels of *menT3* and *menT4* in these conditions. We observed that the transcript levels of *menT3* and *menT4* remained unchanged after being exposed to oxidative stress (Fig. S3b). The survival of parental, Δ*menT3*, Δ*menT4* and Δ*menT4*Δ*T3* strains was comparable upon exposure to either nitrosative or nutritional or acidic stress (Figs. 2b–d and S3e–j). We also compared the ability of wild type, Δ*menT3*, Δ*menT4* and Δ*menT4*Δ*T3* to infect and survive inside THP-1 macrophages. The growth patterns of these strains were similar at days 2, 4 and 6 post-infection in macrophages (Figs. 2e and S4a, b). Taken together, these observations suggest that MenT3 and MenT4 are mutually redundant for in vitro growth, and simultaneous deletion of both *menT3* and *menT4* increased the susceptibility of *M. tuberculosis* to oxidative stress.

## Deletion of both *menT3* and *menT4* impairs the virulence of *M. tuberculosis* in guinea pigs

Previously, we have reported that simultaneous deletion of MazF ribonucleases *mazF3*, *mazF6* and *mazF9* or *higB1* or *vapBC3* or *vapBC4*

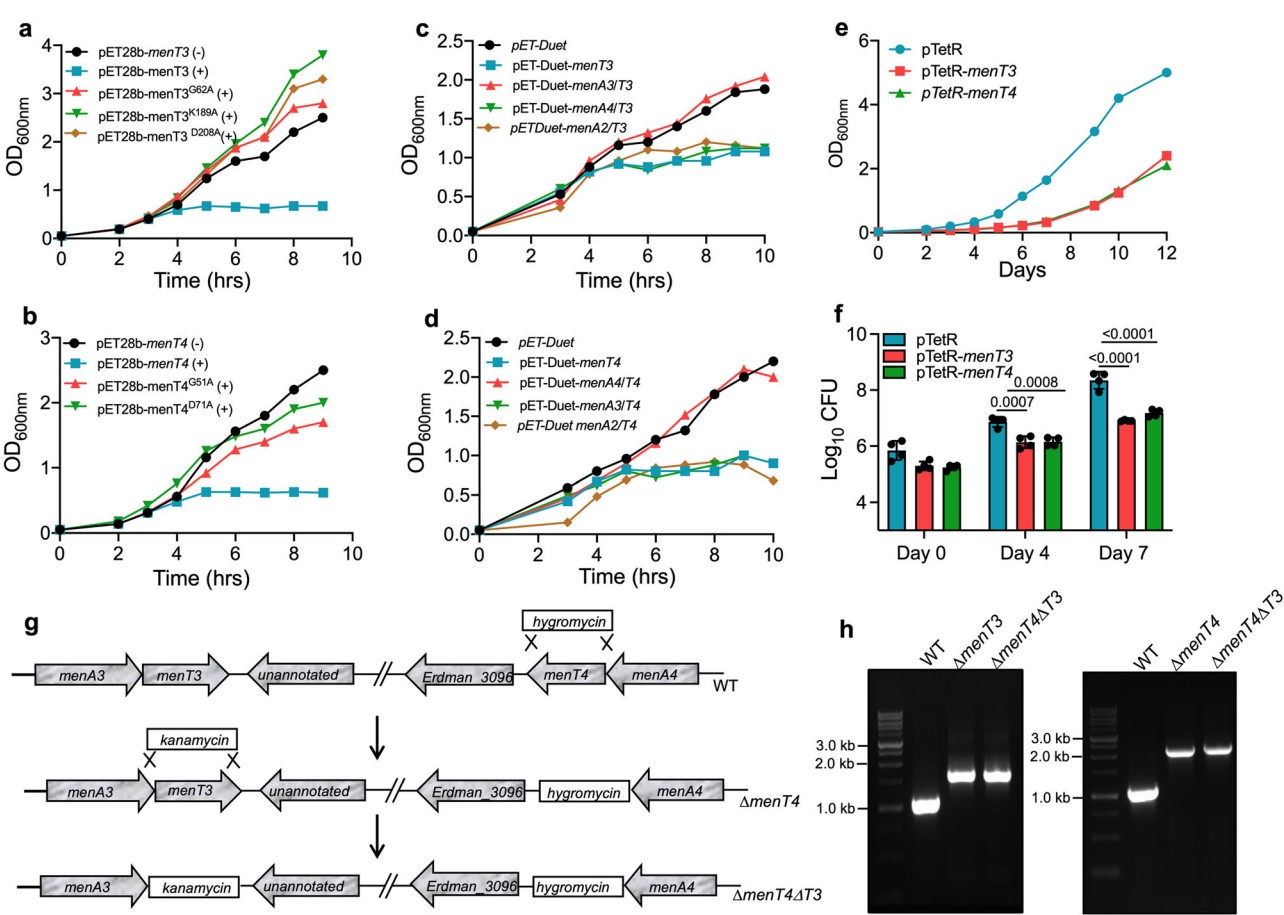

**Fig. 1 | Functional characterization of MenT3 and MenT4 toxins belonging to MenAT subfamily of TA systems from *M. tuberculosis*. a–f** Overexpression of MenT3 and MenT4 inhibits the growth of *E. coli* and *M. tuberculosis*. **a, b** These panels show growth patterns of *E. coli* Bl-21 (pLysS, λDE3) strains harboring pET28b derivatives expressing either wild type or mutant MenT3 (**a**) or wild type or mutant MenT4 (**b**) proteins in the absence or presence of inducer. **c, d** These panels depict growth patterns of *E. coli* BL21 (pLysS, λDE3) strains harboring pET-Duet constructs overexpressing MenT3 (**c**) or MenT4 (**d**) either alone or along with their cognate or non-cognate antitoxins. The growth of various strains was determined by measuring OD600nm. The data shown in these panels are representative of two independent experiments. **e, f** The growth patterns of *M. tuberculosis* H37Rv harboring pTetR derivatives expressing either MenT3 or MenT4 are shown in these panels. The growth of recombinant strains was determined by measuring either OD600nm (**e**) or

bacterial counts (**f**). The data shown in (**e**) is representative of two independent experiments. The data shown in (**f**) is mean ± SD of log10 CFU obtained from two independent experiments, each performed with duplicate cultures. *p* values depicted on the graphs were assessed using one-way ANOVA. **g, h** Construction of Δ*menT4*Δ*T3* strain of *M. tuberculosis*. **g** Schematic representation of *menT3* and *menT4* locus in parental, Δ*menT4* and Δ*menT4*Δ*T3* strain of *M. tuberculosis* Erdman is shown. The open reading frame of *menT4* was replaced with the hygromycin resistance gene in Δ*menT4* strain of *M. tuberculosis*. In the double mutant strain, Δ*menT4*Δ*T3*, the open reading frame of *menT3* and *menT4* were replaced with kanamycin and hygromycin resistance gene, respectively. **h** The replacement of *menT3* and *menT4* with kanamycin and hygromycin resistance gene, respectively, in their respective single and double mutant strain was confirmed by PCR using gene-specific primers. Source Data are provided as a Source Data file.

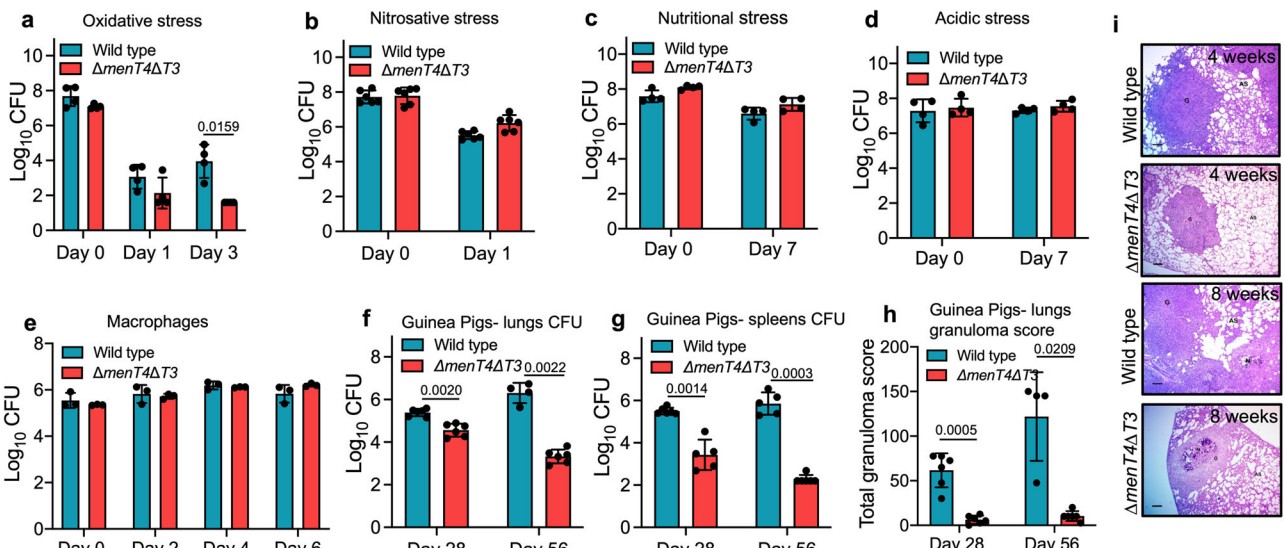

**Fig. 2 | MenT3 and MenT4 are cumulatively essential for *M. tuberculosis* to survive in oxidative stress and establish infection in guinea pigs.** The growth of wild type and Δ*menT4*Δ*T3* was compared after exposure to either oxidative (**a**) or nitrosative (**b**) or nutritional (**c**) or acidic (**d**) stress. The data shown in these panels are mean ± SD of $\log_{10}$ CFU obtained from two (**a**, **c**, **d**) or three (**b**) independent experiments, each performed with duplicate cultures. **e** THP-1 macrophages were infected with various strains, and the number of intracellular bacteria was determined at different time points. The data shown in this panel are mean ± SD of $\log_{10}$ CFU obtained from triplicate wells and representative of two independent experiments performed in duplicates or triplicates. The data shown in these panels in mean ± SD of $\log_{10}$ CFU in lungs (**f**) and spleens (**g**) of guinea pigs (Duncan Hartley strain) infected with either wild type or Δ*menT4*Δ*T3* strain at 4- and 8-weeks post-infection. The data shown in these panels in mean ± SD of $\log_{10}$ CFU obtained from 6 animals (except in (**f**) week 8, wild type $n = 4$ and in (**g**) week 4, Δ*menT4*Δ*T3* $n = 5$ and week 8, wild type $n = 5$). The data shown for the 4-week time point are representative of two independent experiments. The data shown for the 8-week time point are obtained from a single experiment. **h** The data shown in this panel are mean ± SD of total granuloma score in H&E-stained sections of guinea pigs infected with wild type or Δ*menT4*Δ*T3* strain of *M. tuberculosis* at 4- and 8-weeks post-infection. The data shown are obtained from 6 animals from a single experiment (except in week 8, wild type $n = 4$). **i** This panel shows representative images of H&E-stained sections of lung tissues of guinea pigs infected with either wild type or Δ*menT4*Δ*T3* strain at 4- or 8-week post-infection. Scale bar, 100 μm. *p* values depicted on the graphs were assessed using a two-tailed paired *t*-test. Source Data are provided as a Source Data file.

or *vapBC11* or *vapC22* significantly reduced *M. tuberculosis* growth in guinea pigs[21,23,25–27]. However, deletions in either *relE1* or *relE2* or *relE3* or *vapC28* or *vapC21* or *darTG* did not reduce *M. tuberculosis* growth in guinea pigs or mice[21,24,33,34]. In addition to type II TA systems, we have recently shown that MenT2 toxin belonging to the MenAT subfamily is also essential for *M. tuberculosis* pathogenesis in guinea pigs[35]. Previously, high throughput screening assays such as transposon site hybridization (TRASH) and designer array for defined mutant analysis (DeADMAn) have been performed to identify genes necessary for *in vivo* growth of *M. tuberculosis*[36,37]. According to these studies, *menT3* and *menT4* are mutually redundant and not required for *M. tuberculosis* to establish infection in mice[36,37]. In the present study, we compared the growth of wild type and Δ*menT4*Δ*T3* strains in aerosol-infected guinea pigs. We observed that lung bacillary loads were decreased by ~ 6.6- and 950.0-fold in guinea pigs infected with Δ*menT4*Δ*T3* at 4- and 8 weeks post-infection, respectively, relative to parental strain infected guinea pigs (Fig. 2f). As shown in Fig. 2g, we observed ~117.0- and ~3750.0-fold reduction in bacterial loads in spleens of Δ*menT4*Δ*T3* infected guinea pigs relative to wild type strain infected guinea pigs at 4- and 8-weeks post-infection, respectively. In agreement with bacterial burdens, H&E-stained lung sections from guinea pigs infected with wild type strain displayed increased cellular infiltration and severely reduced alveolar spaces (Fig. 2i). This increased cellular infiltration indicates severe inflammation and pathology in lung tissue sections of wild type strain infected guinea pigs at both time points (Fig. 2i). In comparison, the histologically stained lung sections of Δ*menT4*Δ*T3* infected guinea pigs demonstrated intact lung architecture and large alveolar spaces at both time points (Fig. 2i). At both time points, the total granuloma score in lungs of Δ*menT4*Δ*T3* infected guinea pigs was significantly decreased by ~10.0-fold in comparison to total granuloma score in lung sections

from guinea pigs infected with the wild type strain (Fig. 2h). Taken together, we demonstrate that simultaneous deletion of *menT3* and *menT4* results in attenuation of *M. tuberculosis* growth in guinea pigs.

## Deletion of both *menT3* and *menT4* alters the transcriptional profiles of *M. tuberculosis*

We next compared the transcriptional profiles of mid-log phase cultures of wild type and Δ*menT4*Δ*T3* strain to identify the differentially expressed pathways following the simultaneous deletion of *menT3* and *menT4* in *M. tuberculosis*. Using a 2.0-fold cut-off and $p_{adj}$ value of ≤0.05, we observed that compared to the parental strain, transcripts encoding for 36 and 51 proteins were increased and decreased, respectively, in Δ*menT4*Δ*T3* (Supplementary Data 1). The DEGs were further annotated as per the functional category listed by myco-browser (https://mycobrowser.epfl.ch/). Approximately, 33 and 23% of the transcripts with differential expression encoded for either conserved hypothetical proteins or proteins involved in intermediary metabolism and respiration (Table 1). Interestingly, we observed that the expression of toxins belonging to type II TA systems such as Erdman_0269 (Rv0240, *vapC24*), Erdman_0658 (Rv0598c, *vapC27*), Erdman_2181 (Rv1982c, *vapC36*) and Erdman_2744 (Rv2494, *vapC38*) was increased in mid-log phase cultures of Δ*menT4*Δ*T3* relative to the wild type strain (Supplementary Data 1). We also noticed that relative to the parental strain, the levels of transcripts encoding for proteins involved in *M. tuberculosis* adaptation in host tissues were significantly reduced in the Δ*menT4*Δ*T3*. These included Erdman_0097 (Rv0081), a regulatory protein known to be upregulated upon exposure to low oxygen and Erdman_0100 (Rv0084, *hycD*), a protein involved in formate metabolism[38,39]. In addition to these, the transcript levels of Erdman_0321 (Rv0287, *esxG*), a protein involved in the inhibition of phagosome maturation and pathogenesis of *M. tuberculosis*, were also

**Table 1 | Differential expression of genes in ΔmenT4ΔT3 strain relative to the wild type strain**

| S. No | Functional category | No. of down-regulated DEGs | No. of upregu-lated DEGs |
|---|---|---|---|
| 1. | Conserved hypothetical protein | 17 | 11 |
| 3. | Intermediary metabolism and respiration | 9 | 11 |
| 4. | Virulence, detoxification, adaptation | 2 | 4 |
| 5. | PE/PPE | 8 | – |
| 6. | Lipid metabolism | 3 | – |
| 7. | Cell wall and cell processes | 10 | 6 |
| 8. | Regulatory proteins | 1 | – |
| 9. | Information pathways | 1 | 3 |
| 10. | Stable RNA | – | 1 |

The number of upregulated and downregulated genes in the ΔmenT4ΔT3 strain classified as per their functional annotation are shown. The data were obtained from two biological replicates.

reduced in the ΔmenT4ΔT3 strain (Supplementary Data 1)[40]. As shown in Supplementary Data 1, the levels of transcripts encoding for proteins essential for growth (Erdman_3437 (Rv3137), Erdman_3739 (Rv3418c), stress adaptation (Erdman_0630 (Rv0574c), Erdman_2884 (Rv2624c)) and interaction with host Interferon-γ (IFN-γ) (Erdman_1325, Rv1183) were also reduced in ΔmenT4ΔT3 strain relative to the parental strain[36,41–44]. Several studies have reported that replacing an open reading frame with an antibiotic selection marker might affect the expression of neighboring genes due to the polar effect. However, we observed that the relative levels of menT3 and menT4 neighboring genes were comparable in both wild type and ΔmenT4ΔT3 strains (Fig. S5b and Supplementary Data 1). Another possible explanation for the observed attenuated phenotype of ΔmenT4ΔT3 in vivo might be the loss of apolar lipids such as PDIMs during in vitro culturing[45]. In order to rule out this possibility, we compared polar and apolar lipid profiles of mid-log phase cultures of wild type, ΔmenT3, ΔmenT4 and ΔmenT4ΔT3 strains. We observed that the relative levels of apolar lipids (PDIMs, TAG), mycolic acids (MAMEs, FAMEs) and polar lipids (TAG, FAM and DAG) were comparable in wild type and various mutant strains (Fig. S5c). Overall, the data suggests that the attenuated phenotype of the ΔmenT4ΔT3 strain in guinea pigs is most likely associated with reduced levels of transcripts encoding for proteins required for either stress adaptation or virulence of *M. tuberculosis*.

**Deletion of both *menT3* and *menT4* results in attenuation of *M. tuberculosis* growth in mice**

We next compared the growth of wild type and ΔmenT4ΔT3 strain in aerosol-infected Balb/c mice at 4 and 8-weeks post-infection. As shown in Fig. 3a, b, the lung and splenic bacillary loads in ΔmenT4ΔT3 infected mice were decreased by 10.0- and 14.0-fold, respectively, in comparison to wild type infected mice at 4-weeks post-infection. However, at 8 weeks post-infection, we observed 7.0- and 13.0-fold reduction in lungs and splenic bacillary loads in ΔmenT4ΔT3 infected mice compared to mice infected with the wild type strain (Fig. 3a, b). As shown in Fig. S4c, d, we were unable to restore this growth defect in the lungs and spleens of infected animals upon complementation of the double mutant strain with either menT3 or menT4. These observations suggest that both MenT3 and MenT4 contribute cumulatively to the ability of *M. tuberculosis* to establish infection in host tissues. We next performed RNA-seq analysis of lung tissues of uninfected mice and those infected with either parental or ΔmenT4ΔT3 strains at 4-weeks post-infection to understand the plausible underlying mechanisms associated with the attenuated phenotype of the double mutant strain in vivo. Using a $p_{adj}$ value of ≤ 0.05 and a cut-off fold change value of

4.0 and −4.0, we observed that relative to uninfected animals, the expression of 826 and 423 transcripts were increased or decreased in mice infected with the parental strain (Figs. 3c, S4e and Supplementary Data 2). As shown in Fig. 3c, the transcript levels of 312 and 119 genes were increased or decreased, respectively, in mice infected with the ΔmenT4ΔT3 strain relative to uninfected animals (Fig. S4f and Supplementary Data 3). The transcript levels of 330 and 160 genes were increased or decreased in animals infected with ΔmenT4ΔT3 strain in comparison to profiles obtained from parental strain-infected mice (Fig. 3c, d and Supplementary Data 4).

We noticed that relative to the parental strain, the transcript levels of genes encoding for proteins involved in calcium signaling were increased in animals infected with ΔmenT4ΔT3 strain (Fig. 3e and Supplementary Data 4). Previously, it has been shown that increased calcium levels are associated with the induction of numerous antimicrobial pathways such as autophagy and apoptosis[46]. Previously, it has been reported that Adcy1 is activated by *apoA*-1 to promote cholesterol efflux from THP-1 macrophage foam cells[47]. The formation of foamy macrophages is associated with disease progression that leads to cavitation and release of infectious bacilli[48]. In agreement, we observed ∼ 4.0-fold increased expression of *adcy1* in the lungs of mice infected with ΔmenT4ΔT3 strain relative to the parental strain (Supplementary Data 4). *M. tuberculosis* inhibits phagolysosome fusion by interfering with phosphatidylinositol 3-phosphate (PI3P) signaling[49]. We observed increased expression of *adra1a* in lung tissues of ΔmenT4ΔT3 strain-infected mice compared to the parental strain (Supplementary Data 4). Adra1a is involved in the release of inositol 1,4,5-triphosphate and the activation of protein kinase C[50–52]. Studies have shown that deletion of protein kinase C results in higher susceptibility to TB due to increased lung pathology, the release of proinflammatory cytokines and bacterial burdens[53,54]. Numerous studies have shown that bacterial pathogens are able to establish persistent infection by subverting autophagy[55]. We observed that the levels of transcripts encoding for proteins involved in apoptosis and autophagy, such as *slc4a1, bub1, nod1, rnf152, atg16l1, gba, tsc1, phf23, map2k7, mfn2* and *wnt11*, were differentially expressed in lung tissues of mice infected with ΔmenT4ΔT3 strain compared to the parental strain (Fig. 3f and Supplementary Data 4). Autophagy acts as an immune effector mechanism, resulting in phagosomal maturation that mediates mycobacteria clearance[56]. Studies have shown that Bub1 and Phf23 act as negative regulators of autophagy[57–59]. As shown in Supplementary Data 4, we observed reduced expression of *bub1* and *phf23* transcript in lung tissues of ΔmenT4ΔT3 strain infected mice relative to the parental strain infected animals. Additionally, we observed that the levels of transcripts encoding for *atg16l1* were increased in ΔmenT4ΔT3 infected mice lung tissues (Fig. 3f and Supplementary Data 4). Previously it has been shown that depletion of Atg16l1 is associated with increased *M. tuberculosis* growth and susceptibility in mice[60]. The reduced levels of *ptpro* also suggest reduced inflammation and enhanced apoptosis in lung tissues of ΔmenT4ΔT3 infected animals relative to mice infected with the parental strain[61] (Supplementary Data 4). We also observed an increased level of *wnt11* transcript in lung tissues of ΔmenT4ΔT3 strain infected mice compared to the parental strain infected animals (Supplementary Data 4). Previously, it has been shown that overexpression of *wnt11* in intestinal epithelial cells decreased *Salmonella* invasion and inhibited bacteria induced intestinal inflammation[62].

Furthermore, we observed varied expression of several transcripts associated with immune response in ΔmenT4ΔT3 strain infected mice relative to the parental strain infected mice (Supplementary Data 4). We observed reduced levels of transcripts encoding for various cytokines and chemokines such as *il-21, ccl1, cxcl5, cxcl3, ccl4,* and *ccl21a* in ΔmenT4ΔT3 infected mice (Fig. 3g and Supplementary Data 4). Studies have shown that increased expression of Ccl1 is associated with increased endoplasmic reticulum stress and

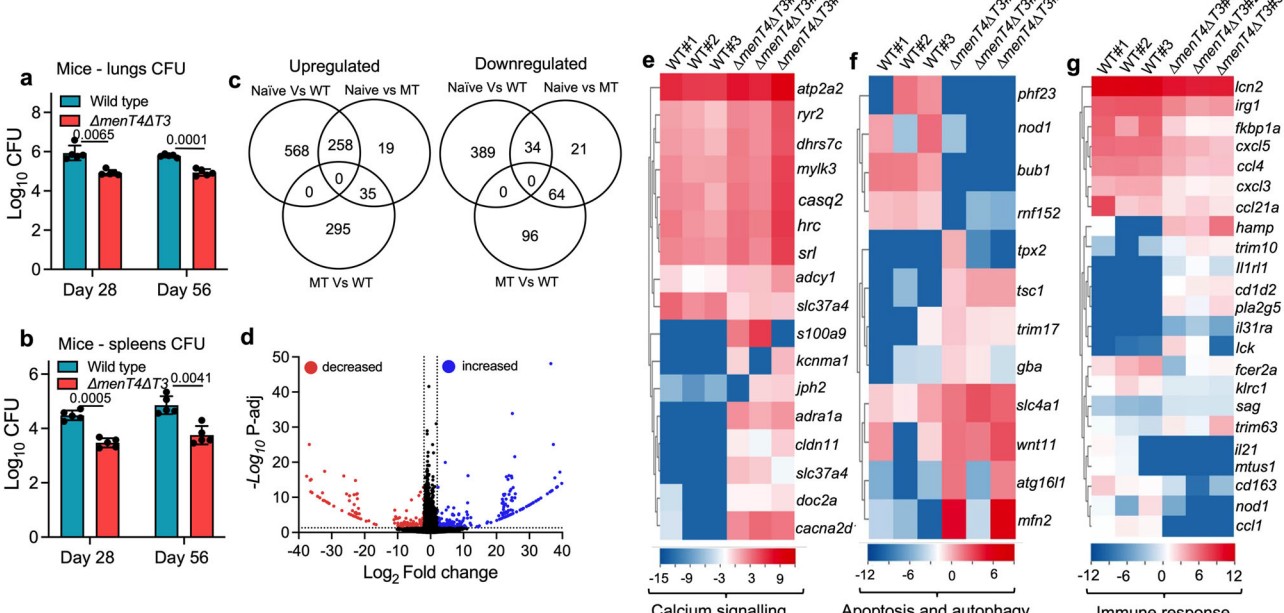

**Fig. 3 | Global transcriptional changes in lung tissues of mice infected with either wild type or $\Delta menT4\Delta T3$ strain at 4 weeks post-infection. a, b** Deletion of *menT3* and *menT4* attenuates *M. tuberculosis* growth in lungs and spleens of mice. The data shown in these panels are mean ± SD of $\log_{10}$ CFU in lungs (**a**) or spleens (**b**) of infected Balb/c mice at 4- and 8-weeks post-infection obtained from 5 animals. The data shown in these panels for 4-week time point are representative of two independent experiments. The data shown for the 8-week time point are obtained from a single experiment. *p* values depicted on the graphs were assessed using a two-tailed paired *t*-test. **c–g** Host transcriptional profiles of lung tissues from uninfected or mice infected with either wild type or $\Delta menT4\Delta T3$ strain. **c** Venn diagram depicting correlation of expression profiles obtained from lung tissues of either uninfected or mice infected with parental or $\Delta menT4\Delta T3$ strain at 4 weeks post-infection. **d** Volcano plot comparing transcription profiles obtained from lung tissues of mice infected with either wild type or $\Delta menT4\Delta T3$ strain at 4 weeks post-infection. The transcripts with increased or decreased expression have been shown as blue or red dots, respectively. The black dots represent the transcripts that remain unchanged and are not statistically different between these two groups. **e–g** Heat maps showing transcripts with differential expression in mice infected with either wild type or $\Delta menT4\Delta T3$ strain at 4 weeks post-infection. The transcripts with differential expression in mice infected with these strains are involved in either calcium signaling (**e**) or apoptosis/autophagy (**f**), or immune response (**g**). The color intensity in heatmaps represents the $\log_2$ value of normalized expression counts. The data shown in (**c–g**) are obtained from three independent biological replicates. Source Data are provided as a Source Data file.

granuloma formation during mycobacterial treatment[63]. Further, it has been reported that elevated plasma levels of Ccl1, Ccl3, Cxcl1, Cxcl9, and Cxcl10 correlate with disease severity in infected individuals[64]. Furthermore, Cxcl5 secreted by pulmonary epithelial cells contributes to excessive neutrophilic inflammation, and mice deficient in Cxcl5 exhibit enhanced survival upon high-dose of *M. tuberculosis* infection compared to wild type mice[65]. The transcript levels of s*ag, hamp, il1rl1, lck, and pla2g5* that encode for proteins involved in the activation of macrophages, neutrophils or T-cells were also increased in lung tissues of mice infected with the mutant strain in comparison to lung tissues from parental strain infected mice (Fig. 3g and Supplementary Data 4)[66–69]. It has also been reported that increased levels of Pla2g5 result in enhanced adaptive immune response and phagocytosis of bacteria by macrophages[70,71]. Taken together, these observations suggest that the in vivo attenuated phenotype of $\Delta menT4\Delta T3$ in lung tissues is most likely associated with increased expression of proteins involved in either calcium homeostasis, apoptosis or autophagy along with decreased expression of transcripts associated with inflammatory response.

### Immunization with $\Delta menT4\Delta T3$ strain imparts protection in mice and guinea pigs against *M. tuberculosis* challenge

Studies have shown that immunization of animals with live attenuated *M. tuberculosis* strains provides long-term protection against challenge with virulent strain as these strains closely mimic the antigenic repertoire of the infectious agent[72–74]. Since *M. tuberculosis* is an intracellular pathogen with pulmonary pathology driven by IFNγ response, C57BL/6 is a widely used strain to study immunological responses, DC-NK crosstalk and $T_H1$ cellular responses[75–80]. Since $\Delta menT4\Delta T3$ was significantly

attenuated for growth in guinea pigs, we next evaluated whether immunization with this strain imparts protection against *M. tuberculosis* challenge in C57BL/6 mice (Fig. 4a). We found that the number of immunizing bacilli (*Mycobacterium bovis* Bacille Calmette-Guerin Pasteur, BCG and $\Delta menT4\Delta T3$) in lungs and spleens of immunized C57BL/6 mice were below the detection limit at 6 weeks post-immunization. In comparison to naive mice, vaccination with $\Delta menT4\Delta T3$ reduced lung and splenic loads of *M. tuberculosis* by ~7.5- and 5.0-folds, respectively, at 4 weeks post-infection (Fig. 4b, c). In comparison to naive mice, immunization with BCG reduced the bacterial counts by 32.0-fold and 7.5-fold in lungs and spleens, respectively, at 4 weeks post-infection (Fig. 4b, c). At 10 weeks post-challenge with *M. tuberculosis*, immunization with $\Delta menT4\Delta T3$ reduced the bacterial numbers by ~4.5-fold and 18.5-fold in lungs and spleens, respectively, in comparison to naive mice (Fig. 4d, e). At 10 weeks post-infection, in comparison to naive mice, immunization with BCG reduced lungs and splenic bacillary loads by ~8.5-fold and 4.4-fold, respectively (Fig. 4d, e). We observed that immunization with $\Delta menT4\Delta T3$ imparted ~4.0-fold increased protection in comparison to immunization with BCG in spleens at 10 weeks post-infection (Fig. 4e).

Next, we assessed the ability of $\Delta menT4\Delta T3$ to impart protection against challenge with *M. tuberculosis* in guinea pigs (Duncan Hartley strain) (Fig. 4f). As observed in C57BL/6 mice, the lung and splenic bacillary loads in BCG and $\Delta menT4\Delta T3$ immunized guinea pigs were below the detection limit at 6 weeks post-immunization. In comparison to naive guinea pigs, we observed ~70.0-fold and 6.0-fold reduction in bacterial numbers in the lungs of $\Delta menT4\Delta T3$ and BCG vaccinated guinea pigs, respectively, at 4 weeks post-infection (Fig. 4g). We observed ~11.0-fold reduction in lung bacillary loads in

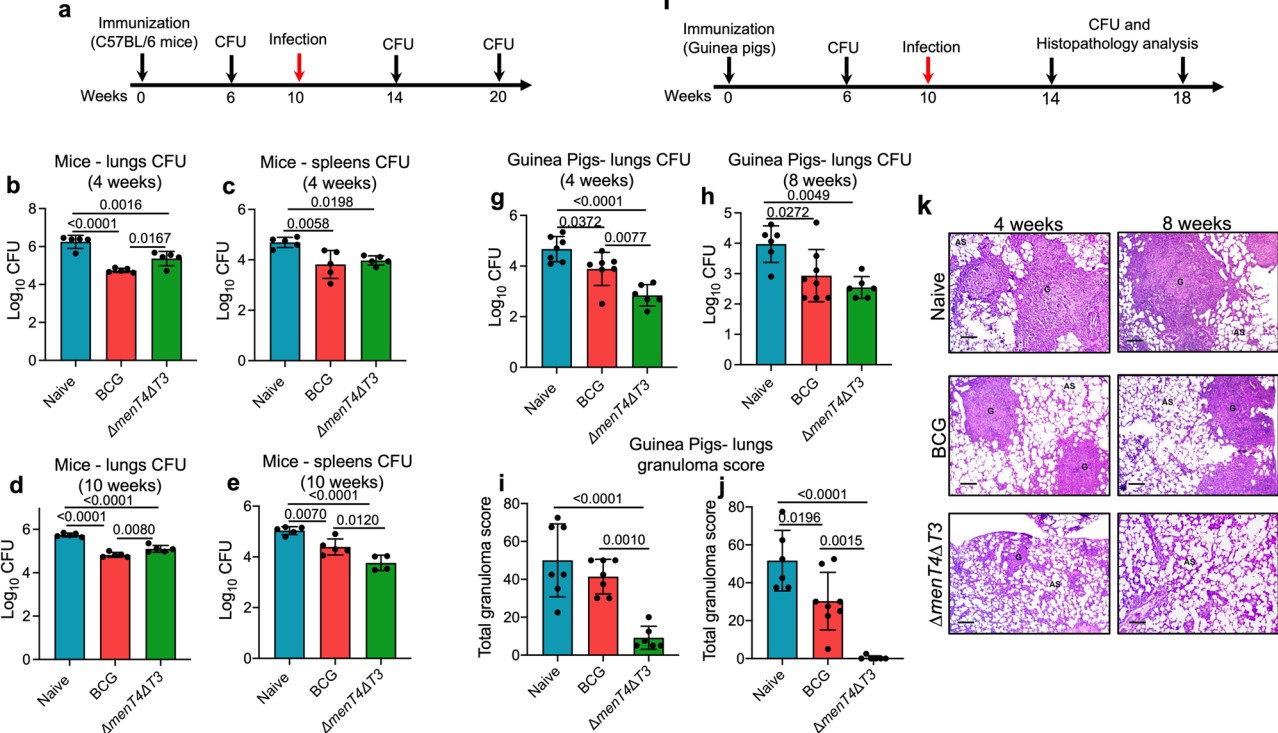

**Fig. 4 | Immunization of mice and guinea pigs with *ΔmenT4ΔT3* imparts protection against *M. tuberculosis* challenge. a** 6–8 weeks old female C57BL/6 mice were immunized subcutaneously with $5 \times 10^5$ CFU of either *M. bovis* BCG or *ΔmenT4ΔT3*. At 10 weeks post-immunization, animals were challenged with *M. tuberculosis* and bacterial enumeration was performed at 4- and 10 weeks post-infection. **b–e** The data shown in these panels is mean ± SD of $\log_{10}$ CFU in lungs (**b**, **d**) and spleens (**c**, **e**) of naive or BCG or *ΔmenT4ΔT3* immunized mice after challenge with *M. tuberculosis* at 4 weeks (**b**, **c**) and 10 weeks (**d**, **e**) post-infection. The data shown in these panels are mean ± SD of $\log_{10}$ CFU obtained from 5 animals from a single experiment (except in (**e**) week 10, *ΔmenT4ΔT3* $n = 4$). **f** 6–8-week-old female guinea pigs (Duncan Hartley strain) were immunized intradermally with $1 \times 10^5$ CFU of either *M. bovis* BCG or *ΔmenT4ΔT3*. At 10 weeks post-immunization, animals were challenged with *M. tuberculosis* and bacterial enumeration and histopathology analysis were performed at 4 and 8-weeks post-infection. The bacterial burdens in the lungs of naive or BCG or *ΔmenT4ΔT3* immunized guinea pigs were determined after challenge with *M. tuberculosis* at 4 (**g**) and 8 weeks (**h**) post-challenge. The data shown are mean ± SD of $\log_{10}$ CFU obtained from 6 animals from a single experiment (except in (**g**) week 4, naïve and BCG $n = 7$ and in (**h**) week 8, BCG $n = 8$). The data shown in this panel are mean ± SD of total granuloma score obtained from H&E stained lung sections of naive or BCG or *ΔmenT4ΔT3* immunized guinea pigs at 4 weeks (**i**) and 8 weeks (**j**) post-challenge. The data shown are obtained from 6 animals from a single experiment (except in (**i**) week 4, naïve and BCG $n = 7$ and in (**j**) week 8, BCG $n = 8$). **k** This panel shows representative images of H&E-stained sections of lung tissues of naive or immunized guinea pigs after infection with *M. tuberculosis* for 4 weeks or 8 weeks. Scale bar, 100 μm. *p* values depicted on the graphs were assessed using one-way ANOVA. Source Data are provided as a Source Data file.

*ΔmenT4ΔT3* immunized animals in comparison to BCG vaccinated guinea pigs (Fig. 4g). Further, we observed immunization with either BCG or *ΔmenT4ΔT3* reduced lung bacillary loads by ~11.0- and 27.0-fold, respectively, in comparison to naive animals at 8 weeks post-challenge (Fig. 4h). However, ~2.5-fold increased protection in *ΔmenT4ΔT3* immunized animals in comparison to BCG immunized guinea pigs was not statistically significant (Fig. 4h). We also performed histopathology analysis of lung sections to determine the extent of disease progression in unvaccinated and vaccinated guinea pigs at both time points. As shown in Fig. 4i, j, the total granuloma score in lung sections of naive guinea pigs was ~50.0 at both time points. In comparison, the total granuloma score in guinea pigs immunized with BCG was reduced by 1.70-fold in comparison to naive animals at 8-weeks post-infection (Fig. 4j). In comparison to naive guinea pigs, immunization with *ΔmenT4ΔT3* significantly reduced the total granuloma score by 5.5- and 124.0-fold at 4- and 8-weeks post-infection, respectively (Fig. 4i, j). The total granuloma score was reduced by 4.5- and 73.0-fold in *ΔmenT4ΔT3* immunized guinea pigs in comparison to guinea pigs immunized with BCG at 4- and 8-weeks post-infection, respectively (Fig. 4i, j). The histopathological analysis revealed large granulomas and significant tissue damage in H&E-stained lung sections from naive animals. In comparison, the number of tubercles and tissue damage was reduced in guinea pigs immunized with BCG (Fig. 4k). In agreement with the total granuloma score, we observed normal lung architecture and few granulomas in the H&E-stained section of guinea pigs immunized with *ΔmenT4ΔT3* at both time points (Fig. 4k). Taken together, these observations suggest that immunization with *ΔmenT4ΔT3* resulted in significant protection against challenge with *M. tuberculosis* in both mice and guinea pigs.

## Immunization of mice with *ΔmenT4ΔT3* induces a $T_H1$-biased response and activated memory T cell response

In order to determine immune correlates of protection in *ΔmenT4ΔT3* immunized mice, we evaluated the antigen-specific adaptive T cell response in spleens of naive, BCG and *ΔmenT4ΔT3* vaccinated C57BL/6 mice at 6 weeks post-immunization (Fig. S6). We observed that immunization of mice with *ΔmenT4ΔT3* promoted antigen-specific $T_H1$ response by increasing the frequency of IFN-γ+ CD4+ T-cells by ~270% and 190% in comparison to naive and BCG immunized mice, respectively (Fig. 5a, b). Also, in comparison to naive mice, immunization with BCG resulted in ~145% increase in the frequency of IFN-γ+ CD4+ T-cells (Fig. 5a, b). Similarly, we observed an increase in the frequency of IFN-γ+ CD8+ T-cells by ~300–450% in *ΔmenT4ΔT3* vaccinated mice compared to naive and BCG immunized mice (Fig. 5d, e). T-bet is the master transcription factor for $T_H1$ cells and also negatively regulates $T_H2$ immune response[81]. In comparison to naive mice, immunization with

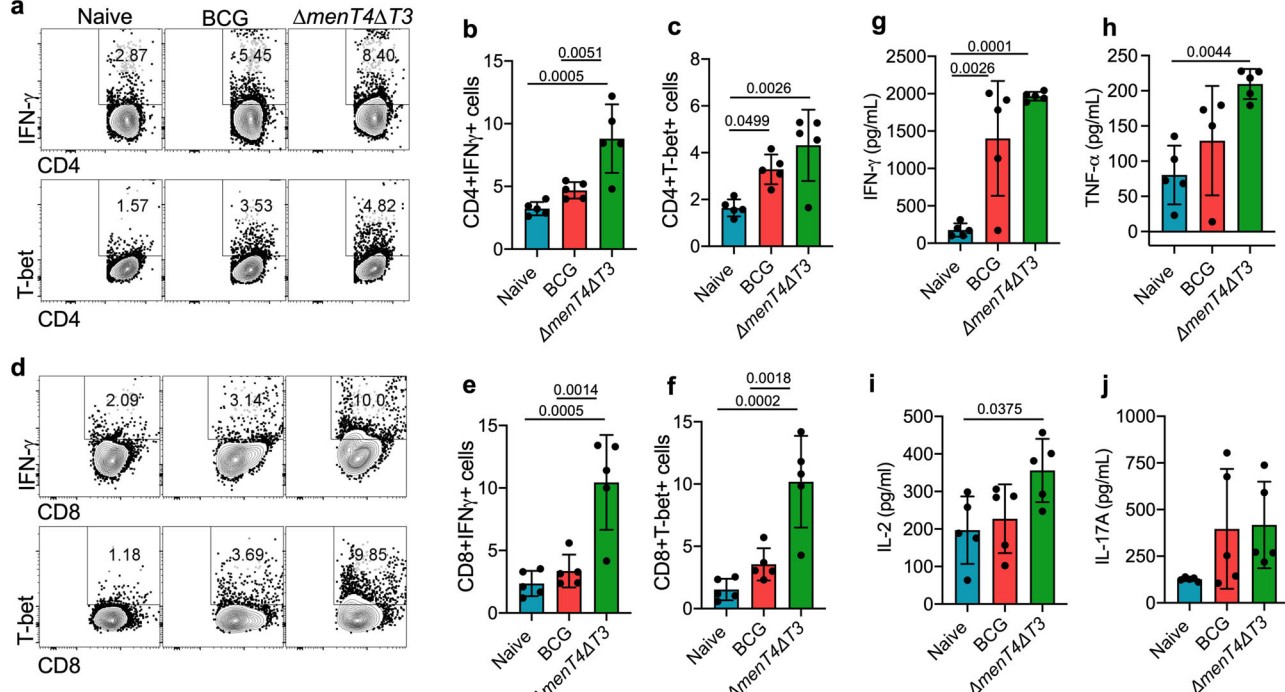

**Fig. 5 | Immunization of mice with ΔmenT4ΔT3 results in increased antigen-specific T$_H$1-response. a–f** C57BL/6 mice immunized with saline or BCG or ΔmenT4ΔT3 were sacrificed at 6 weeks post-immunization, and intracellular cytokines were measured in PPD-stimulated splenocytes. **a** Representative FACS plots depicting percentage frequency of CD4$^+$ IFN-γ$^+$ and CD4$^+$ T-bet$^+$ T cells in spleens of naive or BCG or ΔmenT4ΔT3 immunized mice. These panels show mean ± SD of percentage frequency CD4$^+$ IFN-γ$^+$ (**b**) and CD4$^+$ T-bet$^+$ (**c**) in spleens of naive or BCG or ΔmenT4ΔT3 immunized mice. **d** Representative FACS plots depicting percentage frequency of CD8$^+$ IFN-γ$^+$ and CD8$^+$ T-bet$^+$ T cells in spleens of naive or BCG or ΔmenT4ΔT3 immunized mice. These panels show the percentage frequency of CD8$^+$ IFN-γ$^+$ (**e**) and CD8$^+$ T-bet$^+$ (**f**) T cells in the spleens of naive or BCG or ΔmenT4ΔT3 immunized mice. The data shown in (**b, c, e, f**) are mean ± S.D. of the proportion of T-cells obtained from five animals from a single experiment. The levels of secreted IFN-γ (**g**), TNF-α (**h**), IL-2 (**i**), IL-17A (**j**) were measured in the culture supernatants from PPD-stimulated splenocytes by ELISA. The data shown are mean ± SD of cytokine levels in PPD-stimulated splenocytes obtained from five animals from a single experiment (except in (**h**), BCG $n = 4$). $p$ values depicted on the graphs were assessed using one-way ANOVA. Source Data are provided as a Source Data file.

ΔmenT4ΔT3 significantly enhanced the frequency of T-bet$^+$ expressing CD4$^+$ and CD8$^+$ T cells by ~250% and ~650%, respectively (Fig. 5a, c, d, f). In comparison to BCG immunized mice, the frequency of T-bet$^+$ expressing CD4$^+$ and CD8$^+$ T-cells were increased by 130.0% and 285.0%, respectively, in ΔmenT4ΔT3 immunized mice, respectively (Fig. 5a, c, d, f). We also observed that the frequency of T-bet$^+$ expressing CD4$^+$ and CD8$^+$ T cells increased by 2.0- and 2.31-fold in BCG immunized mice as compared to naive mice (Fig. 5a, c, d, f). The frequency of IL-17A secreting CD4$^+$ T cells in ΔmenT4ΔT3 immunized mice and BCG immunized mice increased by ~150% and 140%, respectively, in comparison to naive mice (Fig. S7a, b). This increase was observed to be non-significant in comparison to the proportion of cells obtained in naive mice. We also determined the levels of various cytokines in the supernatants of PPD-stimulated splenocytes. In agreement with our immunophenotyping experiments, the levels of IFN-γ were significantly increased in culture supernatants from PPD-stimulated splenocytes obtained from BCG and ΔmenT4ΔT3 immunized groups in comparison to naive animals (Fig. 5g). Notably, TNF-α levels were increased by ~1.6-fold and ~2.5-fold in culture supernatants from PPD-stimulated splenocytes obtained from the ΔmenT4ΔT3 immunized group in comparison to BCG immunized and naive group (Fig. 5h). The levels of IL-2 were also significantly increased in culture supernatants from PPD-stimulated splenocytes obtained from ΔmenT4ΔT3 immunized mice in comparison to naive animals (Fig. 5i). In agreement with FACS data, the levels of secreted IL-17A were not significantly changed in supernatants of PPD-stimulated splenocytes from naive, BCG and ΔmenT3ΔT4 immunized mice (Fig. 5j). Previously, it has been shown that the FoxP3 transcription factor is mostly

expressed by regulatory T cells and dampens the antimicrobial inflammatory immune response[82,83]. Immunization of mice with ΔmenT4ΔT3 did not result in significant changes in the frequency of FoxP3$^+$CD4$^+$ T-cells in comparison to naive mice (Fig. S7a, c). These observations suggest that ΔmenT4ΔT3 immunization-induced protection against *M. tuberculosis* challenge might be independent of Treg cells. Together, our data suggests that immunization with ΔmenT4ΔT3 results in the expansion of antigen-specific T$_H$1 response, which may contribute to protection against *M. tuberculosis* infection.

Furthermore, we also evaluated the effector and memory T cell response in spleens of immunized mice (Fig. S8). We observed a significant increase in the frequency of activated memory T helper and T cytotoxic cells by 2.70-fold and 3.80-fold, respectively, in ΔmenT4ΔT3 immunized mice and by ~1.60- and 2.30-fold, respectively, in mice immunized with BCG in comparison to naive mice (Fig. 6a–d). In comparison to BCG immunized mice, the frequency of activated memory CD4$^+$ and CD8$^+$ T cells were increased by ~1.70- and 1.60-fold, respectively, in ΔmenT4ΔT3 immunized mice (Fig. 6a–d). Furthermore, a slight increase, though not statistically significant, in the frequency of effector T helper cells was observed in ΔmenT4ΔT3 immunized as compared to naïve mice (from 14.96 to 17.62%) and BCG immunized mice (from 11.44 to 17.62%) (Fig. S9a, c). Moreover, in comparison to naive mice, the frequency of effector memory CD4$^+$ T cells decreased from 18.2 to 13.26% in ΔmenT4ΔT3 immunized mice (Fig. S9a, d). We also observed that the frequency of effector memory CD4$^+$ T cells was significantly reduced in ΔmenT4ΔT3 immunized mice in comparison to BCG immunized mice (Fig. S9a, d). However, the frequency of effector and effector memory CD8$^+$ T cells were comparable in naïve and

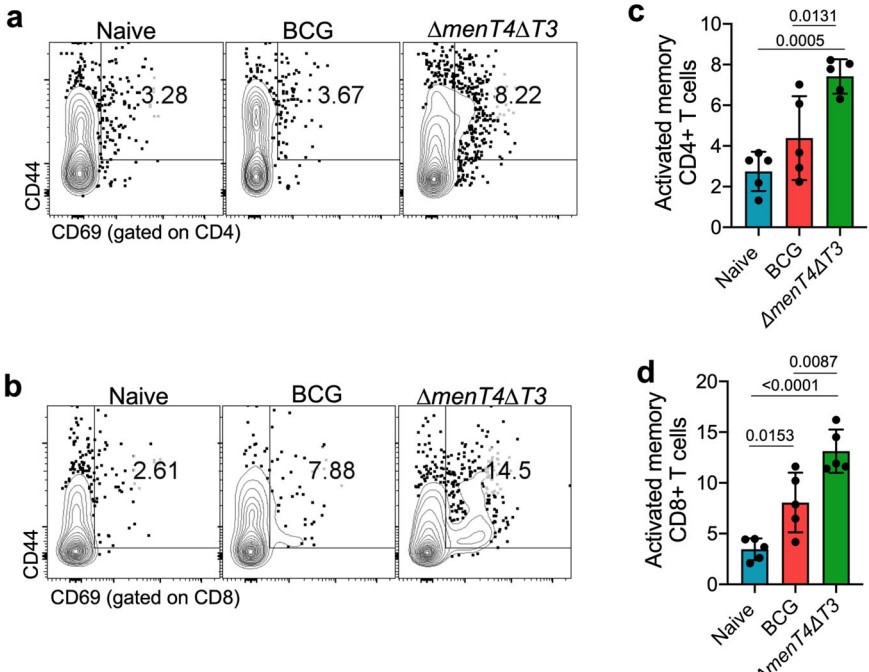

**Fig. 6 | Immunization of mice with *ΔmenT4ΔT3* increases the expansion of activated memory T cell response.** Representative FACS plots showing percentage frequency of CD4⁺ activated memory T_H cells (CD4⁺ CD44⁺ CD69⁺, **a**) and CD8⁺ activated memory T_C cells (CD8⁺ CD44⁺ CD69⁺, **b**) in spleens of naive or BCG or *ΔmenT4ΔT3* immunized C57BL/6 mice. These panels show the percentage frequency of activated memory CD4⁺ T_H cells (**c**) and CD8⁺ T_C cells (**d**) in spleens of naive or BCG or *ΔmenT4ΔT3* immunized mice. The data shown in these panels are mean ± SD of the percentage frequency of cells obtained from five animals from a single experiment. *p* values depicted on the graphs were assessed using one-way ANOVA. Source Data are provided as a Source Data file.

*ΔmenT4ΔT3* immunized mice (Fig. S9b, f, g). In comparison to BCG immunized mice, we observed a significant reduction in the proportion of central memory CD4⁺ and CD8⁺ T cells in *ΔmenT4ΔT3* immunized mice (Fig. S9a, b, e, h). The proportion of central memory CD4⁺ T cells was significantly reduced in *ΔmenT4ΔT3* immunized mice relative to naive mice (Fig. S9a, e). The frequency of effector CD4⁺ T cells, effector memory CD4⁺ and CD8⁺ T cells and central memory CD4⁺ and CD8⁺ T cells were comparable in naive and BCG immunized mice (Fig. S9a–e, g, h). Taken together, we show that immunization of mice with *ΔmenT4ΔT3* strain induces higher antigen-specific T_H1 response and expansion of activated memory T helper and cytotoxic T cell response in comparison to BCG immunized mice. These enhanced immune signatures might be associated with protection imparted by immunization with *ΔmenT4ΔT3* against *M. tuberculosis* challenge.

## Discussion

TA systems are mostly bicistronic genetic elements that are widely distributed in prokaryotes and are involved in stress adaptation, genome maintenance, pathogenesis and control of phage infection. *M. tuberculosis* genome encodes for >90 TA pairs, and most of these belong to either type II or MenAT TA systems[11,12,14]. TA systems belonging to type II subfamily have been extensively characterized in *M. tuberculosis*. However, very limited information is available about MenAT TA systems from *M. tuberculosis*. The three-dimensional structures of MenT1, MenT3 and MenT4 have been solved by X-ray crystallography, but their exact role in *M. tuberculosis* physiology and pathogenesis is still unknown[29–31]. Superimposition of these solved structures revealed that MenT toxins have conserved folds and catalytic sites[30,31]. In agreement with previous reports, we demonstrate that overexpression of MenT3 and MenT4 inhibits *E. coli* and *M. tuberculosis* growth[29,30]. As expected, co-expression of cognate antitoxins or mutation of amino acid residues in the highly conserved motifs abrogated the growth inhibition activity associated with these toxins.

Studies have shown that MenT1, MenT3 and MenT4 homologs from *M. tuberculosis* possess NTase activity in vitro and inhibit protein synthesis by preventing aminoacylation of tRNAs[30,31].

Using temperature-sensitive mycobacteriophages, we generated *M. tuberculosis* mutant strains harboring deletions in either MenT3 or MenT4 or both MenT3 and MenT4. The growth patterns of single and double mutant strains were comparable to the parental strain. The relative abundance and upregulation of a subset of TA systems upon exposure to stress conditions and drugs suggests that these might function in a cumulative manner to enable *M. tuberculosis* to adapt to these conditions[12,21,22]. We observed that MenT3 and MenT4 are mutually redundant, but both MenT3 and MenT4 cumulatively contribute to the adaptation of *M. tuberculosis* upon exposure to oxidative stress. Complementation of the double mutant strain with *menT3* partially restored the growth defect associated with the double mutant strain upon exposure to oxidative stress. We also observed that the relative transcript levels of *menA3*, *menT3*, *menA4* and *menT4* did not significantly change upon exposure to oxidative stress. It has been previously reported that in addition to their own promoter, antitoxins or TA complexes belonging to the type II subfamily can bind to other promoter sequences[84–86]. We hypothesize that by binding to the promoter of genes involved in oxidative stress adaptation, MenAT3 and MenAT4 might regulate their expression. We also demonstrate that the survival of parental, *ΔmenT3*, *ΔmenT4* and *ΔmenT4ΔT3* strains was similar after exposure to either nitrosative stress or nutritional stress or acidic stress.

Several studies have implicated the role of TA systems in the pathogenesis of various microorganisms. It has been shown that TA systems are essential for the virulence of bacterial pathogens such as *H. influenzae*, *V. cholerae*, *S. typhimurium*, *S. aureus*, and *M. tuberculosis*[10]. Here, we report that relative to the parental strain, *ΔmenT4ΔT3* strain was attenuated for growth at both acute and chronic stages of infection in guinea pigs. As observed in the case of

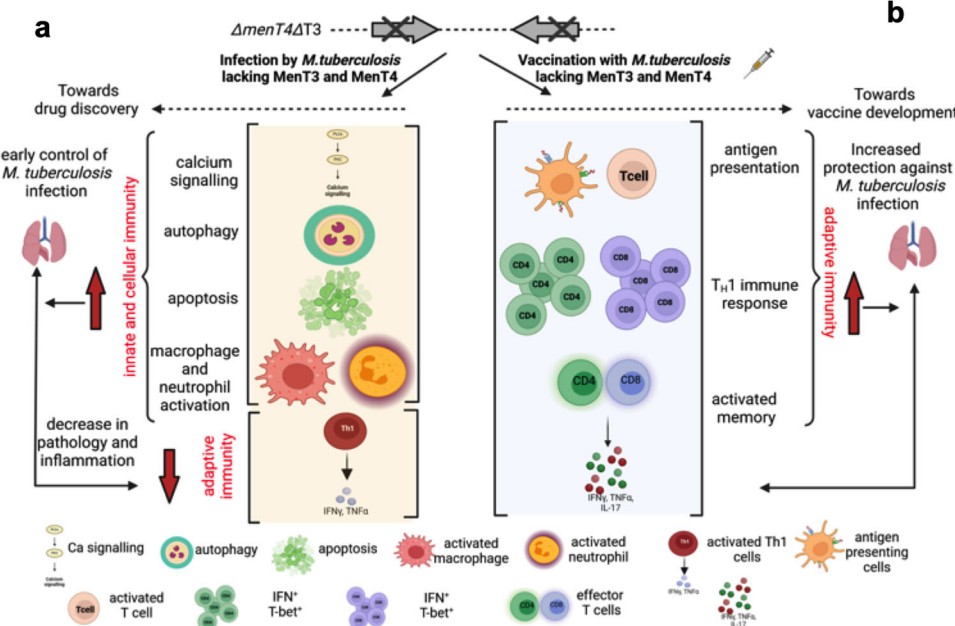

**Fig. 7 | Proposed model for attenuation of *ΔmenT4ΔT3* in host tissues and for protection imparted by *ΔmenT4ΔT3* against *M. tuberculosis*. a** In the present study, we show that deletion of *menT3* and *menT4* impairs the growth of *M. tuberculosis* in mice and guinea pigs. The attenuated phenotype of *ΔmenT4ΔT3* is most likely associated with increased levels of transcripts encoding for proteins involved in calcium signaling, autophagy, apoptosis and activation of macrophages and neutrophils. The increased expression of these pathways results in the generation of a robust innate immune response that restricts *M. tuberculosis* growth and disease progression. **b** Here, we show that administration of *ΔmenT4ΔT3* imparts protection in mice and guinea pigs against challenge with *M. tuberculosis*. The immunization of mice with *ΔmenT4ΔT3* strain results in the expansion of activated memory immune response and induction of antigen-specific $T_H1$ in comparison to naive and BCG immunized mice. This significant increase in the generation of activated memory and $T_H1$ immune response might be associated with protection against *M. tuberculosis* challenge. The figure has been prepared using BioRender.

*ΔmazF3Δ9Δ6* or *ΔvapC22* or *ΔhigB1* or *ΔvapBC3* or *ΔvapBC4* or *ΔvapBC11* or *ΔmenT2*, the growth defect associated with *ΔmenT4ΔT3* strain was more prominent in spleens and during the chronic stage of infection in guinea pigs[21,23,25–27,35]. In agreement with guinea pig data, we observed that the deletion of *menT3* and *menT4* also impaired the growth of *M. tuberculosis* in mice tissues. As expected, we observed significantly reduced tissue damage in lung sections from guinea pigs infected with the double mutant strain in comparison to parental strain infected animals. We also observed that complementation with either *menT3* or *menT4* did not restore the growth defect associated with double mutant strain in host tissues. These findings imply that MenT3 and MenT4 are mutually redundant and that simultaneous deletion of both toxins results in an attenuated phenotype in vivo. However, based on the available data, the possibility of acquisition of secondary site mutation during the generation of *ΔmenT4ΔT3* can't be ruled out. In contrast to the in vivo phenotype, the growth patterns of both parental and *ΔmenT4ΔT3* strains were comparable in macrophages. A possible reason for the lack of growth defect in macrophages could be that the function of MenT3 and MenT4 lies outside macrophages, and these proteins are involved in the interaction or intracellular growth of *M. tuberculosis* inside other host cells such as lung epithelial cells, dendritic cells, adipocytes, neutrophils and mesenchymal stem cells[87–91].

Transcriptomics is widely used to gain a better understanding of the mechanisms for the attenuation of bacterial pathogens[92–94]. Bacterial RNA sequencing revealed that transcript levels of genes encoding for proteins involved in stress adaptation and virulence were reduced in mid-log phase cultures of the double mutant strain as compared to the parental strain. Therefore, we hypothesize that the in vivo attenuated phenotype of the mutant strain might be associated with the reduced expression of these proteins. In order to further unravel the plausible mechanisms associated with the in vivo growth defect of the mutant strain, we also compared the transcriptional profiles obtained from lung tissues of animals infected with either wild type or *ΔmenT4ΔT3* strain. Detailed analysis of the RNA-seq data revealed that the levels of transcripts encoding for proteins involved in calcium signaling, apoptosis and autophagy were increased in *ΔmenT4ΔT3* infected animals relative to parental strain infected animals (Fig. 7a). We also observed that the expression of proteins involved in inflammatory response was reduced in animals infected with the double mutant strain in comparison to parental strain infected animals (Fig. 7a). Taken together, the data suggests that the coordinated execution of these pathways might be associated with the observed growth defect of the mutant strain in host tissues.

Till date, BCG is the only licensed vaccine in use against TB. However, due to its limited efficacy, variable protection and adverse effects, there is an urgent need to develop new vaccine candidates that confer better protection against TB[95–98]. There have been several reports regarding the development of attenuated mutant strains and evaluating their ability to impart protection against *M. tuberculosis* in animal models. The ability of these strains to impart protection is most likely due to the fact that they harbor a complete repertoire of genes encoding for immunodominant antigens. Given that *ΔmenT4ΔT3* was attenuated for growth in guinea pigs and Balb/c mice relative to the parental strain and was being further evaluated as a vaccine candidate, detailed immunological studies were performed in immunized C57BL/6 mice, a preferred model for immune studies. We observed that immunization of C57BL/6 mice with *ΔmenT4ΔT3* significantly reduced the lung bacterial burdens in comparison to naive mice at both 4- and 10 weeks post-infection. The observed protection was similar to the levels seen in BCG-immunized mice. However, the reduction in splenic bacterial counts at 10 weeks post-infection was statistically significant in comparison to bacterial loads observed in BCG-immunized mice. In agreement with mice protection data, we observed that immunization with *ΔmenT4ΔT3* was also able to impart protection against *M.*

*tuberculosis* in guinea pigs. The protection imparted upon immunization with Δ*menT4*ΔT3 at 4 weeks post-infection was significantly enhanced in comparison to protection observed in BCG-immunized guinea pigs. However, the levels of protection were comparable in Δ*menT4*ΔT3 and BCG-immunized guinea pigs at 8 weeks post-challenge. Interestingly, despite similar bacterial loads in lung tissues, analysis of H&E-stained lung sections revealed larger alveolar spaces and minimal cellular infiltration in lung sections of guinea pigs immunized with Δ*menT4*ΔT3 in comparison to BCG immunized animals. The levels of protection observed upon immunization of C57BL/6 and guinea pigs with Δ*menT4*ΔT3 were similar to those reported for other attenuated vaccine candidates such as Δ*leuD* (in guinea pigs) or Δ*panCD* (in guinea pigs) or Δ*RD1*Δ*panCD* (in C57BL/6 mice)[73,99,100].

Several studies have shown that antigen-specific T$_H$1 response is desired to impart protection against *M. tuberculosis* challenge in C57BL/6 mice and guinea pigs[76–78,81,101]. Previous studies have shown that protective immunity against TB depends on an acquired cellular immune response involving T-cell subsets, and a T$_H$1-type response is considered favorable in imparting protection against *M. tuberculosis*[102]. In agreement, we observed significant upregulation of IFN-γ and T-bet expression in CD4$^+$ and CD8$^+$ T cell compartments in Δ*menT4*ΔT3 immunized mice in comparison to BCG-immunized mice. This T$_H$1 skewed response might be associated with the increased protection imparted by Δ*menT4*ΔT3 against *M. tuberculosis* challenge in spleen tissues. Next in order to further understand the immune correlates of protection in the Δ*menT4*ΔT3 immunized mice, we also evaluated the antigen-specific effector and memory T cell response in immunized mice. We observed increased activated memory T cell response in Δ*menT4*ΔT3 immunized mice as compared to BCG immunized or naive mice. Since memory responses are desirable for vaccine-induced long-lasting protection, we speculate that increased activated memory T cell response, as observed in Δ*menT4*ΔT3 immunized mice, might be able to impart protection against relapses as well. It is well documented that T$_H$1 immune responses are necessary for host defence against TB[103–105]. We also observed a significant increase in the secretion of T$_H$1 cytokines such as IFN-γ, TNF-α and IL-2 in culture supernatants of PPD stimulated splenocytes from Δ*menT4*ΔT3 immunized mice, in agreement with the flow cytometry data. Studies have shown that IFN-γ and TNF-α are important for the effective control of *M. tuberculosis* infection[106]. IL-2 has also been shown to stimulate the growth of B-cells, T-cells and NK cells and is essential for cellular immunity and granuloma formation in *M. tuberculosis* infection[106]. We propose that increased amounts of these cytokines in culture supernatants from PPD stimulated splenocytes from Δ*menT4*ΔT3 immunized mice might contribute to its ability to impart protection against *M. tuberculosis* challenge[81,107–110]. However, limitations of the present study include lack of experiments to study immunological responses in guinea pigs, and a small number of animals were utilized in mice experiments.

Overall, this study reveals that although MenT3 and MenT4 are dispensable for in vitro growth, these toxins function in a cumulative manner and are essential for *M. tuberculosis* to establish disease in mice and guinea pigs. We also show that immunization with Δ*menT4*ΔT3 is able to impart protection against *M. tuberculosis* in mice and guinea pigs. We propose that the protection observed in Δ*menT4*ΔT3 immunized animals is most likely associated with increased antigen-specific T$_H$1 biased and activated memory immune response (Fig. 7b). This study also provides the rationale for the development of live vaccines against TB based on the inactivation of virulence-associated pathways regulated by MenT3 and MenT4. Future studies include (1) unmarking and construction of Δ*menT4*ΔT3-based multiple-allele mutant strains (such as *panCD*, *leuD*, *metX*, etc.), and these strains would be evaluated for safety and efficacy studies in compliance with the Geneva consensus[111], (2) identification of small molecule inhibitors against MenT3 and MenT4

proteins and (3) experiments to validate RNA seq data obtained from infected lung tissues.

## Methods

### Ethics approval
Ethics approval for this work was obtained from the Institutional Biosafety Committee and Review Committee on Genetic Manipulation of the Department of Biotechnology, Ministry of Science and Technology, Government of India.

### Bacterial strains, plasmids, and culture conditions
The list of strains and plasmids used in the study are listed in Supplementary Table 1. The list of primers used in the study are listed in Supplementary Table 2. For overexpression studies, genes encoding for MenT3, MenT4 and their point mutants were PCR amplified and cloned in either isopropyl thio-β-galactoside (IPTG)-or anhydrotetracycline (Atc) inducible vectors, pET28b or pTetR, respectively[112]. The recombinant constructs were verified by DNA sequencing. For co-expression studies, the toxin (*menT3* or *menT4*) and antitoxin (*menA2* or *menA3* or *menA4*) were separately cloned into MCS-1 and MCS-2 of the pETDuet vector[113]. The bacterial strains were cultured in either LB broth or LB agar or 7H9 or 7H11 medium as previously described[114].

### Growth inhibition assays in *E. coli* and *M. tuberculosis*
In order to determine whether inducible expression of MenT3 and MenT4 results in growth inhibition of *E. coli*, BL-21 (λDE3, pLysS) was transformed with pET28b or pETDuet derivatives. The expression of proteins in recombinant *E. coli* strains was induced by the addition of 1.0 mM IPTG when OD$_{600nm}$ of 0.4–0.6 was attained. The growth of recombinant strains was assayed by either measuring OD$_{600nm}$ or by spotting diluted cultures on LB agar plates. For growth inhibition studies in *M. tuberculosis*, pTetR or pTetR-*menT3* or pTetR-*menT4* were electroporated in *M. tuberculosis* H$_{37}$Rv. The expression of MenT3 and MenT4 in early-log phase cultures (OD$_{600 nm}$ ˜ 0.2) of recombinant strains was induced by the addition of 50 ng/ml Atc. The growth of parental and recombinant *M. tuberculosis* strains was determined by measuring OD$_{600nm}$ and bacterial numbers at regular intervals. For CFU enumeration, 10.0-fold serial dilutions were prepared and plated on MB7H11 medium at 37 °C for 3–4 weeks.

### Construction of various mutant and complemented strains of *M. tuberculosis*
The single and double mutant strains of *M. tuberculosis* Erdman were generated using temperature-sensitive mycobacteriophages[115]. Briefly, for the construction of Δ*menT4* strain, ˜800 bp upstream and downstream region of *menT4* was PCR amplified and cloned into cosmid vector, pYUB854[115]. The recombinant pYUB854-Δ*menT4* was *Pac* I digested and packaged into phagemid, pYUB159. The recombinant phagemid was electroporated in *M. smegmatis* to generate temperature-sensitive mycobacteriophages. These temperature-sensitive mycobacteriophages were used to transduce mid-log phase cultures of *M. tuberculosis* Erdman (OD$_{600nm}$ ˜ 0.8–1.0) to generate Δ*menT4* strain. For the generation of Δ*menT3* and Δ*menT4*ΔT3 double mutant strain, ˜800 bp upstream and downstream region of *menT3* was PCR amplified and cloned into cosmid vector, pYUB854. The hygromycin resistance gene pYUB854-Δ*menT3* was replaced with the kanamycin resistance gene resulting in pYUB854-Δ*menT3-kan*. The recombinant cosmid, pYUB854-Δ*menT3-kan*, was packaged, and temperature-sensitive mycobacteriophages were prepared as described above. For the construction of Δ*menT3* and Δ*menT4*ΔT3, mid-log phase cultures of *M. tuberculosis* Erdman or Δ*menT4* strain (OD$_{600nm}$ ˜ 0.8–1.0), respectively, were transduced with temperature-sensitive *menT3* mycobacteriophages. The replacement of *menT3* and *menT4* with kanamycin and hygromycin resistance gene, respectively, in Δ*menT4*ΔT3 strain of *M. tuberculosis* Erdman was verified by PCR and

whole genome sequencing. In order to complement the Δ*menT4*Δ*T3* strain, *menA3-menT3* and *menA4-menT4* locus were PCR amplified along with 500 bp upstream region and cloned into pMV306-apramycin. The recombinant construct was electroporated into Δ*menT4*Δ*T3* and transformants were selected on MB7H11 plates containing kanamycin, hygromycin and apramycin.

### Stress experiments

For oxidative stress, early-log phase cultures of various strains ($OD_{600nm} \sim 0.2$) were exposed to 5 mM $H_2O_2$ for either 24 or 72 h. For acidic or nitrosative stress, early log phase cultures were harvested by centrifugation, washed twice and resuspended in MB7H9 medium, pH-5.2 (acidic medium) for 7 days or in the presence of 5 mM $NaNO_2$ (nitrosative stress for 24 h). For nutritional stress, early-log phase cultures were harvested, washed twice and resuspended in 1x tris-buffered saline tween-80 for 7 days. At designated time points, CFU enumeration was performed by plating 10.0-fold serial dilutions on MB7H11 agar plates at 37 °C for 3–4 weeks.

### Macrophage experiments

Human monocyte cell-line THP-1 was obtained from National Centre for Cell Science and cultured in Roswell Park Memorial Institute (RPMI) 1640 medium and differentiated into macrophages by overlaying cells with RPMI medium with 20 ng/ml of phorbol myristate acetate (PMA) for 24 h. For macrophage experiments, differentiated THP-1 cells ($2 \times 10^5$ per well) were infected with various strains at a MOI of 1:10 for 4 h. Subsequently, macrophages were washed with 1x PBS, cultured in RPMI medium containing 200 µg/ml amikacin for 2 h to remove extracellular bacteria and overlaid with RPMI complete medium. The medium was replaced every 48 h during the experiment. At designated time points, macrophages were lysed using 1x PBS containing 0.1% triton X-100. The number of viable bacteria in lysates was determined by plating 10.0-fold serial dilutions on MB7H11 agar at 37 °C for 3–4 weeks.

### Lipid extraction experiments

Briefly, various strains were grown to mid-log phase ($OD_{600nm} \sim 0.8$–1.0). The cultures were harvested by centrifugation and washed twice with 1x PBS. Subsequently, apolar and polar lipids were extracted as previously described[35]. An equal amount of different lipid fractions was loaded on silica plates and resolved using one-dimensional TLC in the different solvent systems as described previously[35]. Different lipids were visualized by staining the TLC with 5% molybdophosphoric acid in ethanol and subsequent charring at 100 °C. TLC images were acquired using a gel documentation system (Bio-Rad).

### Bacterial RNA sequencing

For bacterial RNA-seq experiments, total RNA was isolated from mid-log phase cultures ($OD_{600nm} \sim 0.8$–1.0) of various strains using the Trizol method as previously described[114]. The quality of the isolated RNA was analyzed using Agilent Bioanalyzer. RNA samples were further processed for library preparation and sequenced using the Illumina HiSeq2000 platform. The sequenced reads were aligned to the *M. tuberculosis* Erdman sequence using the Hisat 2 program (Version 2.0.5). The differential gene expression analysis using aligned reads was performed using the Cuffdiff program of the Cufflinks package.

### Animal husbandry

Six to eight weeks old outbred female Duncan Hartley guinea pigs (~250–300 g) were obtained from Disease Free Small Animal House, Lala Lajpat Rai University of Veterinary and Animal Sciences, Hisar. Six to eight weeks old inbred female Balb/c and C57BL/6 mice (~20–25 g) were obtained from the Experimental Animal Facility, NCR Biotech Science Cluster, Faridabad. Animals were housed in a group of either 5

(mice) or 3 (guinea pigs) in individually ventilated cages in BSL-3 labs. The animals were maintained at a room temperature of $22 \pm 3$ °C, relative humidity of 30 to 70%, 15–20 air changes/h, light intensity of 325–350 lux with a 14 h light/10 h dark cycle and noise intensity of <85 db. A sterilized pellet diet (Altromin International, Germany) was offered ad libitum to the animals throughout the experimental period. Ad libitum aqua guard filtered, and autoclaved water was served in bottles fitted with stainless steel nozzles.

### Animal virulence studies

The institutional animal ethics committee of the Translational Health Science and Technology Institute (THSTI) approved the animal experiments. The animal experiments were performed as per the guidelines provided by the committee for the control and supervision of experiments on animals. For aerosol infections, mid-log phase cultures ($OD_{600nm} \sim 0.8$–1.0) of various strains were harvested by centrifugation and washed twice with 1x PBS. Subsequently, single-cell suspensions were prepared, and cultures were diluted to $5 \times 10^8$ CFU (female Balb/c mice) or $1 \times 10^8$ CFU (female Duncan Hartley strain, guinea pigs) in 10 ml saline. The animals were infected using a Glass-col aerosol generation device that implanted ~100 bacilli in lung tissues. For CFU analysis, lungs and spleens were homogenized in saline, 10.0-fold serial dilutions were prepared and plated on MB7H11 agar for 3–4 weeks at 37 °C. For histopathology analysis, lung tissues were fixed in 10% formalin, embedded in paraffin wax and stained with hematoxylin and eosin (H&E). The stained sections were analyzed for tissue damage by a histopathologist. The total granuloma score in H&E-stained lung sections from infected guinea pigs was determined as described earlier[116].

### Host RNA-seq analysis

For host RNA-seq analysis, total RNA was isolated from lung tissues of uninfected Balb/c mice and those infected with either wild type or Δ*menT4*Δ*T3* at 4 weeks post-infection using Qiagen RNA extraction protocol as per manufacturer recommendations[26]. RNA was subjected to DNase I treatment, and the quality and integrity of RNA were analyzed using Agilent Bioanalyzer and subjected to sequencing. The paired-end reads obtained were analyzed for quality control using the NGS QC Toolkit, and those with a Phred score of >Q30 were carried forward for analysis. HISAT2 splice-aware read aligner was used for the alignment of the passed reads to the reference *Mus musculus* (mm10) genome. Stringtie was employed for the assembly and quantification of the aligned reads of all samples. Analysis for differential expression was performed using DESeq2. The heat maps were prepared using TB tools.

### Protective efficacy studies

For immunization experiments, female mice (C57BL/6) or female guinea pigs (Duncan Hartley strain) were immunized with either saline or single-cell suspension containing $5 \times 10^5$ CFU of BCG or Δ*menT4*Δ*T3* strain via subcutaneous route in mice or $1 \times 10^5$ CFU via intradermal route in guinea pigs. For protection studies, at 10 weeks post-immunization, animals were challenged with *M. tuberculosis* H37Rv using a Glass Col aerosol chamber as described above. The protective efficacy imparted by immunization with either BCG or Δ*menT4*Δ*3* was determined by bacterial enumeration and histopathological analysis as described above.

### Flow cytometry experiments

At 6 weeks post-immunization spleens from naive and immunized C57BL/6 mice were isolated and passed through a 70 µm cell strainer to obtain a single-cell suspension. Subsequently, RBCs were lysed with ice-cold ammonium–chloride–potassium (ACK) lysis buffer (20–30 s at room temperature) and resuspended in RPMI complete media. For immunophenotyping experiments, splenocytes were seeded in a 96-

well plate ($2 \times 10^5$ cells per well). The splenocytes were stimulated with 10 µg/ml purified protein derivative (PPD) for 72 h. Thereafter, cells were harvested, washed and incubated with anti-CD16/CD32 Fc block for 15 min. The cells were stained for surface markers with Live-Dead dye-BV421, anti-mouse: CD45.2-APC-Cy7, anti-CD4-PerCp-Cy5.5, anti-CD8-PE, anti-CD62L-APC, anti-CD44-PE-Cy7 and anti-CD69-FITC, for 30 min at 4 °C (BD Biosciences, Supplementary Table 3). For intracellular staining of cytokine and transcription factor, single cells of PPD-stimulated splenocytes were additionally treated with Golgi stop containing Brefeldin A (BD Biosciences) for 6 h before the completion of the incubation as previously described[117,118]. Following this, cells were harvested, washed, incubated for 20 min with anti-CD16/CD32 Fc block and stained with Live-Dead dye-Cy7, anti-CD4-PerCp-Cy5.5 and anti-CD8-BV510 for 1 h. Subsequently, intracellular staining of cells was performed using a cytofix/cytoperm kit with anti-IFN-γ-PE, anti–IL-17A-PE-Cy7, anti-T-bet-APC and anti-Foxp3-BV421 (Supplementary Table 3)[119]. The stained samples were washed twice with ice-cold 1x PBS, resuspended in FACS staining buffer and acquired on BD Canto II (BD Biosciences). The acquired data were analyzed using Flow Jo (Treestar) software version X.

### ELISA experiments
Splenocytes from immunized C57BL/6 mice were seeded at a density of $2 \times 10^5$ cells per well in a 96-well plate and stimulated with 10 µg/ml PPD for 72 h. The levels of IFN-γ, TNF-α, IL-2 and IL-17A in supernatants of PPD-stimulated splenocytes were measured by ELISA as per the manufacturer's recommendations.

### Statistical analysis
The statistical tests and graphs were prepared using GraphPad Prism (Version 9.5.1). The statistical tests used for data analysis are mentioned in the respective figure legends. The statistical analysis of RNA-seq data was performed using the Negative Binomial Wald test. *$p < 0.05$, **$p < 0.01$, ***$p < 0.001$ and ****$p < 0.0001$ were considered statistically significant.

### Reporting summary
Further information on research design is available in the Nature Portfolio Reporting Summary linked to this article.

## Data availability
The RNA-seq data generated in the study have been deposited in NCBI-SRA repositories under accession code Bioproject PRJNA997775 for *Mus musculus* and PRJNA997818 for *M. tuberculosis*. Source data are provided with this paper.

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

## Acknowledgements

This work was supported by the DBT/Wellcome Trust India Alliance Fellowship (IA/S/19/2/504646) awarded to R.S. R.S. is a recipient of the Ramalingaswami fellowship (BT/HRD/35/02/18/2009) and the National Bioscience award (BT/HRD/NBA/37/01/2014). The authors acknowledge the staff members of the BSL-3 facility and experimental animal facility, THSTI for technical help during animal and BSL-3 experiments. The authors also acknowledge the experimental animal facility, NCR Biotech Science Cluster, Faridabad and disease free small animal house, LUVAS for providing mice and guinea pigs, respectively. T.P.G. and S.C. acknowledge DBT and CSIR for their respective research fellowship. The authors sincerely thank late Dr. Ashok Mukherjee for the histopathology analysis. Ms. Neha Rana and Ms. Saruchi Wadhwa are acknowledged for their help with cloning experiments. The authors acknowledge Agrigenome and Bionivid for bacterial and host RNA sequencing, respectively. The authors acknowledge Dr. Amar Deep, Dr. Deepak Saini and Dr. Sheetal Gandotra for scientific discussions. The authors sincerely thank Dr. Bhabatosh Das, THSTI, for sequencing the genomic DNA of *M. tuberculosis* strains. Mr. Rajesh, Mr. Sher Singh and Ashish are acknowledged for technical assistance.

## Author contributions

R.S. conceived the idea and supervised the experiments. T.P.G., S.C., N.K.C. and S.K. performed in vitro and stress experiments. T.P.G., S.C. and N.K.C. performed animal virulence studies. T.P.G. and S.C. performed animal protection studies. T.P.G. and S.C. analyzed bacterial and host RNA-seq data. Z.A., S.C. and T.P.G. performed immune response studies. K.G.T. supervised growth inhibition experiments, A.A. supervised immune response studies. R.S. and T.P.G. wrote the manuscript with inputs from other authors.

## Competing interests

The authors declare no competing interests.
