## [Peer Review File · Nature Communications]

REVIEWER COMMENTS

Reviewer #1 (Remarks to the Author):

In the current study, Gosain et al. investigated the physiological effects of deleting two *M. tuberculosis* toxins from the toxin-antitoxin systems, MenT3 and MenT4, on stress responses and pathogenesis. The authors further evaluated the protective ability of a MenT3-MenT4 mutant in an animal model. Reports on this family of toxins are valuable because they have only recently been reported in *M. tuberculosis* and their involvement in disease progression is still not understood. This study is the first to link this family of genes to the pathogenesis of tuberculosis. However, there are some major points to be considered before assuming this link.

Major points:

- Double mutation: The main shortcoming of this paper is the authors' decision to delete both MenT3 and MenT4 without fully comprehending their targets and physiological functions. What was the rationale behind the deletion of both toxins in a single strain? Is there any evidence of functional interactions or shared targets? Other toxins of the same family appear to have functionally distinct targets and possible downstream effects (eg. Members of the most studied toxin family in *M. tuberculosis*, the VapBC family, are surprisingly specific to particular tRNA isoforms, whereas others digest ribosomal RNA, suggesting that they might function independently). If MenT3 and MenT4 act in independent pathways, simultaneous deletion may exacerbate physiological effects that may never occur together.
- Lack of full complementation: No experiment included full complementation by reintroducing MenT3 and MenT4 simultaneously to the double mutants, which is crucial to eliminate any technical artifacts and validate that the observed phenotype is a result of gene deletions. In fact, the results indicate that phenotype restoration was not obtained with a single toxin complementation, thus not ruling out that the sensitivity to oxidative stress and softer pathogenesis could be collateral damage resulting from technical flaws (such as non-specific off-target mutations, the presence of hygR cassette or other genes from the phage vector, deletion of unannotated flanking genes, etc.). I believe that full complementation is essential for obtaining accurate conclusions from these experiments.

Minor points:

- Very little is described in the paper regarding their targets, supposedly because of the lack of studies on these toxins. In fact, it was only very recently that a new report by Pierre Genevaux's group showed the activity, albeit weak, of MenT4 on several tRNAs. I suggest adding more information about the possible targets in the introduction, cautiously minding that the very few published studies on mycobacteria involve target identification by in vitro assays, which have been reported to be non-specific.
- Figure 2E shows similar growth patterns in macrophages, despite the observed susceptibility to oxidative stress. How can this result be reconciled with the known oxidative stress in the phagosome of macrophages?

- To promote data sharing and reuse, it is recommended that Supplementary Tables S3 and S4 be replaced with a complete list of detected genes and their expression levels/q-values from the authors' experiments, rather than just the genes that passed their filtering criteria.

- Where it reads "...we observed that the expression of toxins belonging to Type II TA systems such as Erdman_0269, Erdman_0658, Erdman_2181 and Erdman_2744 were increased in mid-log phase cultures...", I recommend including the toxin names or close homologs, along with their locus tag IDs, for straightforward identification.

- The RNA-seq data could have been further explored and discussed. Was there any specific transcript signature for the MenT3/MenT4 mutants? Isn't this general transcriptomic profile expected for any attenuated pathogen due to the decreased lung damage – eg. lower proinflammatory cytokine levels? Accordingly, Figure 3 shows a significant change in the histological composition of the lesions (smaller granulomas in the double mutants). This means that there are considerably fewer activated lymphoid- and myeloid-derived cells in the tissue compared to the granuloma-dominated lung tissue. The distinct cellular subpopulations by itself may explain the transcriptomic differences. Although not the main scope of this study, ideally, experiments such as this could provide much richer information if the experimental design involves spatial transcriptomic techniques to visualize the local transcriptomic changes inside the important subpopulations (for example, focusing inside the granuloma). The heterogeneity of the surrounding tissue and the drastic differences in cell composition introduce considerable noise in bulk RNA-seq, making it more difficult to draw meaningful conclusions.

- Where it reads: "we demonstrate that overexpression of MenT3 and MenT4 inhibits E. coli and M. tuberculosis growth in a bacteriostatic manner" – The authors have not determined whether it has a bacteriostatic or bactericidal effect, and it could be either one, especially when ectopically expressed.

- On page 20: "Previously, it has been reported that overexpression of toxins belonging to Type IV TA systems altered the colony morphology and resulted in the formation of lemon shaped cells". I do not understand the reasoning behind this sentence. A phenotype is largely determined by the toxin mechanism of action within the cells. By simply sharing the same TA system classification (an artificial classification solely based on the nature of the toxin and antitoxin and their inhibiting interactions), why would one expect them to exhibit the same phenotype? As noted previously, even toxins from the same family may display divergent phenotypes. Instead, the phenotype should be comparable to that of other toxins that share the same target or mode of action, regardless of the TA type.

- The attenuation phenotype of the toxins has not been largely discussed in comparison with the few other studies that have performed high-throughput mutagenesis screening (such as Tn-seq) in animal models of M. tuberculosis and M. bovis. Have MenT3/T4 mutants been identified as hits in these previous studies? Alternatively, have MenT3 and MenT4 ever shown to be upregulated in any disease condition or oxidative stress experiment (eg. RNA-seq datasets)? This would strengthen the link between these toxins and their response to stress and disease establishment.

- The authors failed to discuss how their double mutant compared with the other attenuated strains that have already been proposed as vaccine options, such as the classic panCD, leuCD, RD region deletion strains developed by Bill Jacobs's lab. It was not clear whether the authors are proposing MenT3 and MenT4 deletion mutants as a possible viable, safer vaccine alternative (I would then argue there is not enough data for that assumption), menT3 and menT4 as an additional set of genes to delete in those already attenuated strains, or a third option.

Suggestions for figures

- Some of the Figure 1 y-axis labels appear to be missing.
- Figure S3A groups are indistinguishable (lacks color legend)
- For better data visualization without the need to refer to the legend text constantly, I would suggest labeling the plots on top of each panel (for example, in Fig 2, adding "Oxidative Stress" to the top of panel A, "Nitrosative stress" to panel B, and so on. The same can be done for animal model experiments, identifying the tissue that corresponds to each plot). This increases the readability of the plot and makes it simple and obvious at first glance.
- Please include all three biological replicates in the RNA-seq heatmaps and normalize the raw read counts as RPKM or TPM for better comparison. In addition, please adjust the color map range according to your data. In the way it is presented, it is impossible to have a blue color in the heatmap, since it is impossible to have negative log₂ raw read counts – it can be misleading to think that none of those transcripts are downregulated.

Reviewer #2 (Remarks to the Author):

Review of "Mycobacterium tuberculosis strain with deletions in menT3 and menT4 is attenuated and confers protection in mice and guinea pigs" by Gosain and others.

This work describes the development of a double mutant (dm) strain of Mtb that overexpresses two toxins from the type IV TA systems, from the MenAT subfamily. A comprehensive description of the molecular biology used to create and test the dm is provided; there is also a fairly comprehensive testing of the immunogenicity and protective efficacy associated with de facto immunization studies. Development of a dm for this purpose (as a vaccine, or at the very least a powerful tool to study Mtb and the necessity to gain protection) is laudable. Further, the comprehensive nature of the description of the generation of this dm is appreciated. What is difficult, however, is to decipher the contribution of this work within a single title (and submission). Although the entire paper is topical under the aegis of Mtb, there is just too much competition between the expert molecular biology and creation and prelim in vitro testing of this dm, and the subsequent in vivo testing showing reduced pathology grossly, and cell kinetics via flow cytometry, etc. – there is a feeling of much too much for one manuscript; to be lumped under one umbrella actually dilutes the overall story.

Before undergoing a complete review, authors are strongly encouraged to split this work into at least two

separate submissions; one that details and defines the importance of MenAT overexpression as a function of the mutant in *E. coli* inhibition, for instance. Segregation will allow the first manuscript to provide adequate license to the subsequent submission that will include details of the pathogenicity and physiological response as it relates to the two rodent models used in these studies.

Reviewer #3 (Remarks to the Author):

Manuscript #NCOMMS-23-36208

"Mycobacterium tuberculosis strain with deletions in menT3 and menT4 is attenuated and confers protection in mice and guinea pigs" by Gosain et al

The genome of *M. tuberculosis* encodes more than 90 toxin-antitoxin genes, but whether these genes are all transcribed into functional products, and what roles these products play in the general biology and pathogenesis of *M. tuberculosis* is yet unknown. The authors attempt to address this gap by examining the role of toxin-antitoxin MenT3 and MenT4 by creating menT3 and menT4 deficient *M. tuberculosis*. Overall, the authors demonstrate that MenT3/T4 play a role in combating oxidative stress, and in promoting growth in vivo. Further, the authors demonstrate that these attenuated *M. tuberculosis* Δ menT4 Δ T3 mutant can be employed as a vaccine that can provide enhanced protection compared to the gold standard BCG in guinea pigs in terms of reduction of bacterial burdens in organs and reduction in lung inflammation. Benefits of the Δ menT4 Δ T3 vaccine were less readily observed in mice, though the authors devote much effort into characterizing the cellular immune responses in mice to the Δ menT4 Δ T3 vaccine construct. The authors also draw conclusions about the mechanism of enhanced protection based on their findings, though these conclusions are not supported by direct follow-up studies. Overall, the authors provide important new information about the role of this toxin-antitoxin pair in the in vitro and in vivo lifestyles of *M. tuberculosis* and provide encouraging data that the Δ menT4 Δ T3 mutant could be useful as a novel vaccine candidate.

General Comments:

1. Review of the manuscript would be made easier by the addition of line numbers to the body of the manuscript
2. The manuscript is generally well written, though could use some minor edits for grammar and syntax.
3. Several sections of the manuscript could be shortened and some of the descriptive information about published data moved to the Discussion. In particular, beginning with the paragraph at the end of page 14 and continuing through page 15, much of this information could be condensed.

Major Comments

4. The authors do not provide important information pertaining to animal studies such as the strain (for the guinea pigs) and origin of the animals, housing conditions, etc. Additionally, the authors did not explicitly state how many animals were used in any of the experiments, or how many times each of these experiments were reproduced.

5. The authors variably use C57BL/6 and Balb/c mice for different studies (primary infection versus vaccination-challenge studies) and should provide a rationale why they switched between these strains given that these strains vary in their inherent susceptibility to *M. tuberculosis*.

6. In the Materials and Methods section, the authors describe RNASeq experiments for bacterial RNASeq and host RNASeq. However, the authors do not provide important details such as how many culture replicates were used for bacterial RNASeq studies, how many animals were used for host RNASeq studies, or how any experimental replicates may have been incorporated into either data analyses. The authors should also consider using an FDR cutoff for their RNASeq DEG comparisons rather than p-value.

7. The authors devote a substantial effort into characterizing immune responses elicited by vaccination with Δ menT4 Δ T3 in mice and trying to link these immune features as correlates of protection. However, data presented in Fig.4 demonstrate that in mice, protection was either worse (Fig. 4B), similar, or moderately improved (Fig. 4E) with the Δ menT4 Δ T3 vaccine. Though fewer reagents for cell characterization of guinea pigs are available, it would have been more appropriate to characterize responses in the animal model where enhanced protection was observed. Any data taken from the mice may be difficult to extrapolate to the guinea pig.

Minor Comments

8. In the Materials and Methods section, the authors describe flow cytometry and ELISA experiments yet do not state which animal species these experiments are being performed on. This should be clarified.

9. On page 13, the final sentence of the first paragraph, there may be a typo. The sentence states "...reduced levels of transcripts encoding for proteins required for either in vitro growth of stress...". The authors did not demonstrate that the Δ menT4 Δ T3 mutations impact growth in vitro, so perhaps they meant in vivo?

10. The authors make several attempts to complement the Δ menT4 Δ T3 double mutation, yet in some circumstances cannot. While this could be due to circumstances proposed by the authors, the authors can not rule out that this is also indicative of a second site mutation in the mutant and should discuss this.

11. Vaccination and challenge experiments presented in Figure 4 represent data from mice and guinea pigs, yet when these experiments are described in the text (starting on the top of page 16), it is unclear to the reader that the first set of experiments was performed in mice (and in which mouse strain). This should be clarified in the text.

12. On page 21, the authors conclude that because the Δ menT4 Δ T3 mutant did not have a growth defect in macrophages, that macrophages are unable to mimic the conditions encountered by *M. tuberculosis* inside host tissues. This is a strained interpretation given that there are many examples of mutants of *M. tuberculosis* that are attenuated for growth in macrophages and that growth defect is directly linked to attenuation in vivo. A better interpretation may be that the functions of MenT4 and MenT3 lie outside of growth in macrophages, which still does not rule out the importance of these factors for interactions with many other host cells and aspects of the host environment.

13. In the Discussion, that authors propose a model for the proposed biological mechanism for the in vivo phenotypes of the Δ menT4 Δ T3 mutant. While these are reasonable hypotheses, putting them into such a figure imparts to the reader that many aspects of this model have been directly addressed in these studies with the Δ menT4 Δ T3 mutant, such as the role of altered calcium signaling, changes in the Th1 responses etc, and they have not. I might suggest not including this figure.

Response to the queries raised by reviewers

We thank the reviewers for their careful assessment of our manuscript. We found their comments and suggestions useful, which enabled us to improve the manuscript. In the revised manuscript we have addressed most of the comments raised by the reviewers. We have also rewritten a few sections in the revised manuscript for better clarity. In the responses below, the reviewers' comments are in bold and our responses are indented in italics. We sincerely hope that the revised manuscript will now be considered suitable for publication in Nature Communications.

Reviewer's comments

Reviewer #1 (Remarks to the Author):

In the current study, Gosain et al. investigated the physiological effects of deleting two M. tuberculosis toxins from the toxin-antitoxin systems, MenT3 and MenT4, on stress responses and pathogenesis. The authors further evaluated the protective ability of a MenT3-MenT4 mutant in an animal model. Reports on this family of toxins are valuable because they have only recently been reported in M. tuberculosis and their involvement in disease progression is still not understood. This study is the first to link this family of genes to the pathogenesis of tuberculosis. However, there are some major points to be considered before assuming this link.

We thank the reviewer for appreciating the work done and the careful evaluation of the manuscript. We found the suggestions to be useful. We have addressed most of the concerns raised by the reviewers and also rewritten a few sections of the manuscript for better clarity. We sincerely hope that the revised manuscript will be considered suitable for publication in Nature Communications.

Major points:

Double mutation: The main shortcoming of this paper is the authors' decision to delete both MenT3 and MenT4 without fully comprehending their targets and physiological functions. What was the rationale behind the deletion of both toxins in a single strain? Is there any evidence of functional interactions or shared targets? Other toxins of the same family appear to have functionally distinct targets and possible downstream effects (e.g. Members of the most studied toxin family in *M. tuberculosis*, the VapBC family, are surprisingly specific to particular tRNA isoforms, whereas others digest ribosomal RNA, suggesting that they might function independently). If MenT3 and MenT4 act in independent pathways, simultaneous deletion may exacerbate physiological effects that may never occur together.

*The genome of *M. tuberculosis* encodes for a huge repertoire of TA systems suggesting that these modules might function in a cumulative manner. Most of these TA systems belong to Type II subfamilies, such as 50 VapBC, 10 MazEF, 3 HigBA, and 3 RelBE TA systems. Recent studies have shown that functional redundancy exists between TA systems. For example, we had earlier reported that MazF ribonucleases contribute synergistically to *M. tuberculosis*'s ability to adapt to conditions such as oxidative stress and nutrient deprivation (Tiwari et al., 2015). In another study by Winther et al., it has been reported that VapC toxins cleave tRNA specifically within the*

anticodon loop (Winther et al., 2016). In the same study, it has been shown that functional redundancy exists between VapC toxins, and few of them cleave the same substrates. In addition to Type II TA systems, *M. tuberculosis* genome encodes for four homologs of MenAT TA systems. The toxins (MenT₁₋₄ belonging to the MenAT subfamily) are characterized by the presence of conserved nucleotidyl transferase (NTase), like domain annotated as the domain of unknown function (DUF) 1814. MenT proteins harbor four highly conserved motifs, including the nucleotidyl transferase (NTase) like domain. Motifs I and II are present at the amino-terminus and comprise of hG[G/S]_{x9-13}DhD domain. Due to its similarity to RxxRxxR seen in tRNA NTases, motif III, KLxAaxxR is predicted to be involved in base stacking interactions for incoming nucleotides. Motif IV comprises a pentad of conserved amino acids +DxxD. The three-dimensional structures of MenT1, MenT3 and MenT4 toxins from *M. tuberculosis* have been solved at a resolution of 1.65 Å, 1.6 Å and 1.2 Å, respectively (Cai et al., 2020; Xu et al., 2023). These toxins feature a common toxin fold and are bilobed globular proteins. A more detailed analysis revealed that the overall architectures of MenT3 and MenT4 are similar with a root mean square deviation (RMSD) of 4.7 Å (Cai et al., 2020). Upon superimposition of the two structures, the authors observed that the active site residues of MenT3 (D80, K189 and D211) and MenT4 (D69, K171 and D186) were at a similar position (Cai et al., 2020). Superimposition of MenT1 with MenT3 or MenT4 resulted in an RMSD of 3.829 Å and 4.232 Å, respectively, and a similar alignment of core regions (Xu et al., 2023). It has also been reported that MenT1, MenT3 and MenT4 possess nucleotidyl transferase activity (Cai et al., 2020; Xu et al., 2023). MenT3 displays a preference for pyrimidines and modifies *M. tuberculosis* tRNA^{Ser} isoacceptors (Cai et al., 2020). Cai et al., also demonstrated that MenT3 weakly modifies tRNA^{Leu}. Recently, it has been reported that MenT4 exhibits a preference for GTP and modifies several tRNAs, including tRNA^{Ser} (Xu et al., 2023). Taken together, these findings suggest that MenT3 and MenT4 overexpression inhibits bacterial growth by preventing aminoacylation and tRNA charging. We have included this information in the revised manuscript.

Lack of full complementation: No experiment included full complementation by reintroducing MenT3 and MenT4 simultaneously to the double mutants, which is crucial to eliminate any technical artifacts and validate that the observed phenotype is a result of gene deletions. In fact, the results indicate that phenotype restoration was not obtained with a single toxin complementation, thus not ruling out that the sensitivity to oxidative stress and softer pathogenesis could be collateral damage resulting from technical flaws (such as non-specific off-target mutations-, the presence of hygR cassette or other genes from the phage vector, deletion of unannotated flanking genes, etc.). I believe that full complementation is essential for obtaining accurate conclusions from these experiments.

We understand the concern raised by the reviewer. However, on account of the limited availability of genetic tools for generating fully complemented strains, we are unable to perform the requested experiment. The problem is further accentuated by the lack of availability of 4 useable selection markers in the field. We have already used three selection markers (hygromycin, kanamycin and apramycin) for the complementation of double mutant strain with either menT3 or menT4. We understand the concern being raised and acknowledge that this is a limitation of the study. In the present study, we have used temperature-sensitive mycobacteriophages to create knock-out strains. This method, developed in the laboratory of Dr. Bill Jacobs, is one of the most widely used

approaches to generate mutant strains. The deletion of *menT3* and *menT4* in the genome of *M. tuberculosis* was confirmed by both PCR and whole-genome sequencing. Whole genome sequencing revealed that sequences aligning to *menT3* and *menT4* were missing from the reads obtained from Δ *menT4* Δ *T3* genomic DNA in comparison to reads obtained from the wild type strain. As shown in Fig. S2, the sequencing reads obtained from both strains mapped to *menT3* and *menT4* neighbouring genes. Several studies have shown that the replacement of an open reading frame with an antibiotic selection marker might result in a polar effect on the expression of neighbouring genes. To rule out this possibility, we have performed qPCR studies and reported that the relative levels of *menT3* and *menT4* neighbouring genes were comparable in the wild type and Δ *menT4* Δ *T3* double mutant strain. Also, we have included a table with a complete list of detected genes in RNA-seq experiments and their respective p-values and q-values (Table S3). Another possibility for the observed attenuated phenotype could be the loss of virulence-associated lipids such as PDIMs upon *in vitro* culturing. As shown in the original manuscript, we observed that the lipid profiles of wild type and Δ *menT4* Δ *T3* were comparable. As suggested by the reviewer, we have also included data for the characterization of single mutant strains in the revised manuscript. We show that the growth of wild type, Δ *menT3* and Δ *menT4* strains of *M. tuberculosis* was comparable in different stress conditions and in THP-1 macrophages. In the present study, we show that simultaneous deletion of both *menT3* and *menT4* resulted in increased susceptibility of *M. tuberculosis* upon exposure to oxidative stress. However, the survival of wild type, Δ *menT3* and Δ *menT4* strains were comparable in oxidative stress conditions. Also, relative to the parental strain, infection of guinea pigs or mice with Δ *menT4* Δ *T3* resulted in attenuated growth in lungs and spleens. High throughput screening assays such as transposon site hybridization (TRASH) and designer array for defined mutant analysis (DeADMan) have been performed to identify genes necessary for *in vivo* growth of *M. tuberculosis* (Sassetti et al., 2003; Lamichhane et al., 2005). According to these studies, *MentT3* and *MentT4* are not required for *M. tuberculosis* to establish infection in mice (Sassetti et al., 2003; Lamichhane et al., 2005). Based on these observations, we conclude that both *MentT3* and *MentT4* are mutually redundant and a phenotype is observed upon simultaneous deletion of both *menT3* and *menT4* in *M. tuberculosis*. We have also included this additional data and information in the revised manuscript.

Minor points:

Very little is described in the paper regarding their targets, supposedly because of the lack of studies on these toxins. In fact, it was only very recently that a new report by Pierre Genevoux's group showed the activity, albeit weak, of *MentT4* on several tRNAs. I suggest adding more information about the possible targets in the introduction, cautiously minding that the very few published studies on mycobacteria involve target identification by *in vitro* assays, which have been reported to be non-specific.

We thank the reviewer for this suggestion. As suggested, we have included this information in the introduction section of the manuscript.

Figure 2E shows similar growth patterns in macrophages, despite the observed susceptibility to oxidative stress. How can this result be reconciled with the known oxidative stress in the phagosome of macrophages?

*We thank the reviewer for this comment. We agree with the reviewer that intracellular pathogens are exposed to oxidative stress inside macrophages. However, the actual levels of oxidative stress experienced by bacteria in liquid cultures and in macrophages may vary depending on various factors, such as the activation state of the macrophage, duration and amount of oxidants. Also, in addition to macrophages, it has been reported that *M. tuberculosis* is able to infect other cell types such as lung epithelial cells, dendritic cells, adipocytes, neutrophils, and mesenchymal stem cells (Lyadova, 2017, Scordo et al., 2016, Khan et al., 2017, Beigier et al., 2017, Mihret, 2012). Since the growth patterns of wild type and Δ menT4 Δ T3 were comparable in macrophages, there is a possibility that these proteins are involved in interaction with other host cells that are involved in host-pathogen interaction. We have included this information in the revised manuscript.*

To promote data sharing and reuse, it is recommended that Supplementary Tables S3 and S4 be replaced with a complete list of detected genes and their expression levels/q-values from the authors' experiments, rather than just the genes that passed their filtering criteria.

As suggested by the reviewer, we have provided the list of detected genes as Table S3, Table S4, Table S5 and Table S6 in the revised manuscript.

Where it reads “...we observed that the expression of toxins belonging to Type II TA systems such as Erdman_0269, Erdman_0658, Erdman_2181 and Erdman_2744 were increased in mid-log phase cultures...”, I recommend including the toxin names or close homologs, along with their locus tag IDs, for straightforward identification.

As suggested, we have included this information in the revised manuscript.

The RNA-seq data could have been further explored and discussed. Was there any specific transcript signature for the MenT3/MenT4 mutants? Isn't this general transcriptomic profile expected for any attenuated pathogen due to the decreased lung damage – eg. lower proinflammatory cytokine levels? Accordingly, Figure 3 shows a significant change in the histological composition of the lesions (smaller granulomas in the double mutants). This means that there are considerably fewer activated lymphoid- and myeloid-derived cells in the tissue compared to the granuloma-dominated lung tissue. The distinct cellular subpopulations by itself may explain the transcriptomic differences. Although not the main scope of this study, ideally, experiments such as this could provide much richer information if the experimental design involves spatial transcriptomic techniques to visualize the local transcriptomic changes inside the important subpopulations (for example, focusing inside the granuloma). The heterogeneity of the surrounding tissue and the drastic differences in cell composition introduce considerable noise in bulk RNA-seq, making it more difficult to draw meaningful conclusions.

We agree with the reviewer that the transcriptional profiles observed in lung tissues of mice infected with the Δ menT4 Δ T3 strain could be attributed to the attenuated phenotype of the double mutant strain. In our host RNA-seq data, we observed increased and decreased expression of transcripts encoding for proteins involved in either calcium homeostasis/apoptosis/autophagy and immune response, respectively, in lung tissues of Δ menT4 Δ T3 infected animals relative to the

parental strain infected animals. Future experiments have been planned to validate these findings. As of now, we don't have the capability to perform single-cell RNA-seq. We thank the reviewer for this valuable comment.

Where it reads: “we demonstrate that overexpression of MenT3 and MenT4 inhibits E. coli and M. tuberculosis growth in a bacteriostatic manner” – The authors have not determined whether it has a bacteriostatic or bactericidal effect, and it could be either one, especially when ectopically expressed.

We thank the reviewer for this comment. We have modified this statement in the revised manuscript.

On page 20: “Previously, it has been reported that overexpression of toxins belonging to Type IV TA systems altered the colony morphology and resulted in the formation of lemon shaped cells”. I do not understand the reasoning behind this sentence. A phenotype is largely determined by the toxin mechanism of action within the cells. By simply sharing the same TA system classification (an artificial classification solely based on the nature of the toxin and antitoxin and their inhibiting interactions), why would one expect them to exhibit the same phenotype? As noted previously, even toxins from the same family may display divergent phenotypes. Instead, the phenotype should be comparable to that of other toxins that share the same target or mode of action, regardless of the TA type.

We thank the reviewer for this comment. We have deleted this statement from the revised manuscript.

The attenuation phenotype of the toxins has not been largely discussed in comparison with the few other studies that have performed high-throughput mutagenesis screening (such as Tn-seq) in animal models of M. tuberculosis and M. bovis. Have MenT3/T4 mutants been identified as hits in these previous studies? Alternatively, have MenT3 and MenT4 ever shown to be upregulated in any disease condition or oxidative stress experiment (eg. RNA-seq datasets)? This would strengthen the link between these toxins and their response to stress and disease establishment.

High throughput screening assays such as transposon site hybridization (TRASH) and designer array for defined mutant analysis (DeADMAN) have been performed to identify genes necessary for in vivo growth of M. tuberculosis (Sasseti et al., 2003; Lamichhane et al., 2005). According to these studies, MenT3 and MenT4 are not required for M. tuberculosis to establish infection in mice (Sasseti et al., 2003; Lamichhane et al., 2005). In the present study, we show that relative to the parental strain, infection of guinea pigs or mice with $\Delta menT4\Delta T3$ resulted in attenuated growth in both lungs and spleens. Based on these observations, we conclude that both MenT3 and MenT4 are mutually redundant and a phenotype is observed upon deletion of both menT3 and menT4 in M. tuberculosis. In agreement with previous studies, we also observed that menT3 and menT4 transcript levels remain unchanged after being exposed to oxidative stress conditions (Namouchi et al., 2016). It has been previously reported that antitoxins are cleaved by cellular proteases in stress conditions and this results in TA system activation. We hypothesize, that the relative protein levels of toxins and antitoxins belonging to menAT3 or menAT4 alter and regulate bacterial growth upon exposure to oxidative stress. It has also been demonstrated that in addition

to their own promoter, Type II TA systems bind to other DNA sequences (Wen et al., 2018; Guo et al., 2019; Sun et al., 2017). Therefore, we hypothesize that in addition to their native promoter, MenAT3 and MenAT4 might bind to other promoters and regulate the expression of genes involved in adaptation to oxidative stress.

The authors failed to discuss how their double mutant compared with the other attenuated strains that have already been proposed as vaccine options, such as the classic panCD, leuCD, RD region deletion strains developed by Bill Jacobs's lab. It was not clear whether the authors are proposing MenT3 and MenT4 deletion mutants as a possible viable, safer vaccine alternative (I would then argue there is not enough data for that assumption), menT3 and menT4 as an additional set of genes to delete in those already attenuated strains, or a third option.

*In the present study, we show that immunization with Δ menT4 Δ T3 imparts better protection against *M. tuberculosis* challenge in guinea pigs compared to mice. In comparison to naive guinea pigs, we observed ~ 70.0-fold and 28.0-fold reduction in lung bacillary loads in Δ menT4 Δ T3 immunized animals at 4- and 8-weeks post-challenge, respectively. We also observed that the lung bacterial loads were reduced by 6.0-fold and 11.5-fold in BCG-immunized guinea pigs at 4- and 8 weeks post-immunization relative to the naïve group. The levels of protection seen in Δ menT4 Δ T3 immunized guinea pigs is slightly better compared to the protection afforded by immunization of guinea pigs with either Δ leuD or Δ panCD strain. It has been shown that guinea pigs immunized with either Δ leuD or Δ panCD showed a reduction of ~ 10.0-fold in lung bacterial loads compared to naive animals (Sampson et al., 2004). It has been reported that subcutaneous immunization of mice with Δ panCD reduced lung bacterial loads by ~ 5.0-fold at 28 days post-challenge (Sambandamurthy et al., 2002). In another study, it was shown that subcutaneous immunization of C57BL/6 mice with Δ RD1 Δ panCD significantly reduced lung bacterial loads by 10.0-fold relative to the naïve group. The levels of protection observed in these studies were comparable to those observed in Δ menT4 Δ T3 immunized mice (Sambandamurthy et al., 2006). Future experiments would include the generation of multiple-allele mutant strains, including Δ menT4 Δ T3, to be evaluated for safety and efficacy in compliance with the Geneva consensus (Kamath et al., 2005). We propose that attenuated strains with deletions of menT3 and menT4 have the potential to be explored further as a vaccine candidate. As suggested, we have included this information in the revised manuscript.*

Suggestions for figures

Some of the Figure 1 y-axis labels appear to be missing.

As suggested, we have incorporated these labels in the revised figure.

Figure S3A groups are indistinguishable (lacks color legend) - For better data visualization without the need to refer to the legend text constantly, I would suggest labeling the plots on top of each panel (for example, in Fig 2, adding “Oxidative Stress” to the top of panel A, “Nitrosative stress” to panel B, and so on. The same can be done for animal model experiments, identifying the tissue that corresponds to each plot). This increases the readability of the plot and makes it simple and obvious at first glance.

We apologise for this. We thank the reviewer for this comment. As suggested, we have incorporated these suggestions in the revised figure for better readability.

Please include all three biological replicates in the RNA-seq heatmaps and normalize the raw read counts as RPKM or TPM for better comparison. In addition, please adjust the color map range according to your data. In the way it is presented, it is impossible to have a blue color in the heatmap, since it is impossible to have negative log₂ raw read counts – it can be misleading to think that none of those transcripts are downregulated.

We thank the reviewer for this comment. In the revised manuscript, we have identified the differentially expressed transcripts using a P_{adj} -value of ≤ 0.05 and log₂ fold change of 4.0 and -4.0. We have replotted the heat maps using the normalized read counts obtained from three biological replicates. Based on this reanalysis, we have rewritten the results obtained from host RNA-seq experiments in the revised manuscript.

Reviewer #2 (Remarks to the Author):

Review of “Mycobacterium tuberculosis strain with deletions in menT3 and menT4 is attenuated and confers protection in mice and guinea pigs” by Gosain and others. This work describes the development of a double mutant (dm) strain of Mtb that overexpresses two toxins from the type IV TA systems, from the MenAT subfamily. A comprehensive description of the molecular biology used to create and test the dm is provided; there is also a fairly comprehensive testing of the immunogenicity and protective efficacy associated with de facto immunization studies. Development of a dm for this purpose (as a vaccine, or at the very least a powerful tool to study Mtb and the necessity to gain protection) is laudable. Further, the comprehensive nature of the description of the generation of this dm is appreciated. What is difficult, however, is to decipher the contribution of this work within a single title (and submission). Although the entire paper is topical under the aegis of Mtb, there is just too much competition between the expert molecular biology and creation and prelim in vitro testing of this dm, and the subsequent in vivo testing showing reduced pathology grossly, and cell kinetics via flow cytometry, etc. – there is a feeling of much too much for one manuscript; to be lumped under one umbrella actually dilutes the overall story.

*We thank the reviewer for appreciating the amount of work presented in the manuscript. In the revised manuscript, we have included data concerning the characterization of Δ menT3 and Δ menT4 strains of *M. tuberculosis*. Based on the results obtained, we conclude that both MenT3 and MenT4 are mutually redundant and a phenotype is observed upon simultaneous deletion of both menT3 and menT4. We have also rewritten a few sections of the manuscript for better clarity. We sincerely hope that the revised manuscript will be considered suitable for publication in *Nature Communications*.*

Reviewer #3 (Remarks to the Author):

Manuscript #NCOMMS-23-36208

"Mycobacterium tuberculosis strain with deletions in menT3 and menT4 is attenuated and confers protection in mice and guinea pigs" by Gosain et al

The genome of *M. tuberculosis* encodes more than 90 toxin-antitoxin genes, but whether these genes are all transcribed into functional products, and what roles these products play in the general biology and pathogenesis of *M. tuberculosis* is yet unknown. The authors attempt to address this gap by examining the role of toxin-antitoxin MenT3 and MenT4 by creating menT3 and menT4 deficient *M. tuberculosis*. Overall, the authors demonstrate that MenT3/T4 play a role in combating oxidative stress, and in promoting growth in vivo. Further, the authors demonstrate that these attenuated *M. tuberculosis* Δ menT4 Δ T3 mutant can be employed as a vaccine that can provide enhanced protection compared to the gold standard BCG in guinea pigs in terms of reduction of bacterial burdens in organs and reduction in lung inflammation. Benefits of the Δ menT4 Δ T3 vaccine were less readily observed in mice, though the authors devote much effort into characterizing the cellular immune responses in mice to the Δ menT4 Δ T3 vaccine construct. The authors also draw conclusions about the mechanism of enhanced protection based on their findings, though these conclusions are not supported by direct follow-up studies. Overall, the authors provide important new information about the role of this toxin-antitoxin pair in the in vitro and in vivo lifestyles of *M. tuberculosis* and provide encouraging data that the Δ menT4 Δ T3 mutant could be useful as a novel vaccine candidate.

*We thank the reviewer for appreciating the amount of work presented in the manuscript. In the revised manuscript, we have included data concerning the characterization of Δ menT3 and Δ menT4 strains of *M. tuberculosis*. Based on the results obtained, we conclude that both MenT3 and MenT4 are mutually redundant and a phenotype is observed upon simultaneous deletion of both menT3 and menT4. We have also rewritten a few sections of the manuscript for better clarity. We sincerely hope that the revised manuscript will be considered suitable for publication in Nature Communications.*

General Comments:

Review of the manuscript would be made easier by the addition of line numbers to the body of the manuscript.

As suggested, we have added the line numbers to the body of the revised manuscript.

The manuscript is generally well written, though could use some minor edits for grammar and syntax.

We thank the reviewer for this comment. We have checked for grammatical errors and also rewritten a few sections of the manuscript for better clarity and readability.

Several sections of the manuscript could be shortened and some of the descriptive information about published data moved to the Discussion. In particular, beginning with the paragraph at the end of page 14 and continuing through page 15, much of this information could be condensed.

As suggested, we have rewritten few sections of the manuscript for better readability and clarity.

Major Comments

The authors do not provide important information pertaining to animal studies such as the strain (for the guinea pigs) and origin of the animals, housing conditions, etc. Additionally, the authors did not explicitly state how many animals were used in any of the experiments, or how many times each of these experiments were reproduced.

*As suggested, the information pertaining to animal studies, such as the strain, origin of the animals and housing conditions, have been included in the revised manuscript. The virulence studies in guinea pigs have been performed twice. In the first experiment, the growth of wild type and $\Delta menT4\Delta T3$ was compared at 4 weeks post-infection. In the second experiment, the growth of both wild type and $\Delta menT4\Delta T3$ was determined at 4- and 8 weeks post-infection. We have included the data obtained from experiment 2 in the manuscript (Fig. 2F and 2G). Also, the virulence studies in mice have been performed twice. In the first experiment, the growth of wild type and $\Delta menT4\Delta T3$ was compared at 4- and 8 weeks post-infection. We have included this data in the manuscript (Fig. 3A and 3B). In the second experiment, the growth of wild type, $\Delta menT4\Delta T3$, $\Delta menT4\Delta T3:menT3$ (*menT3* complemented), $\Delta menT4\Delta T3:menT4$ (*menT4* complemented) was compared in mice at 4 weeks post-infection. This data has been shown in Fig. S4C and S4D of the revised manuscript. The vaccine efficacy experiments in mice and guinea pigs have been performed once. We have included these results in Fig. 4 in the revised manuscript. We have included the number of animals used in each experiment in the figure legend. If the reviewer insists, we can include results from the virulence experiment in guinea pigs (experiment 1) as supplementary data.*

The authors variably use C57BL/6 and Balb/c mice for different studies (primary infection versus vaccination-challenge studies) and should provide a rationale why they switched between these strains given that these strains vary in their inherent susceptibility to *M. tuberculosis*.

*The influence of genetic background on the outcome of cellular and humoral immune responses to vaccines is well documented. This difference in the genetic makeup makes C57BL/6 mice suitable for studying Th1 cell-based pathology, and this pathology is driven by the generation of IFN γ response against intracellular pathogens. Moreover, previous studies have also shown that C57BL/6 mice are suitable for optimal DC–NK cross-talk and enhanced cellular responses (Zeng et al., 2016). Since *M. tuberculosis* is an intracellular pathogen with pulmonary pathology driven by IFN γ mediated responses, C57BL/6 is a widely used mouse strain for immunogenicity studies in the context of *M. tuberculosis* (Silva et al., 2022, Weinrich et al., 2001, Hanna et al., 2021, Khan et al., 2021, Jia et al., 2022).*

In the Materials and Methods section, the authors describe RNASeq experiments for bacterial RNASeq and host RNASeq. However, the authors do not provide important details such as how many culture replicates were used for bacterial RNASeq studies, how many animals were used for host RNASeq studies, or how any experimental replicates may have

been incorporated into either data analyses. The authors should also consider using an FDR cutoff for their RNASeq DEG comparisons rather than p-value.

As suggested, we have provided the details for the number of replicates for bacteria and host RNA-seq experiments. Also, we have included FDR-cutoff (P-adj) values in the RNAseq data sheets (Table S3, Table S4, Table S5 and Table S6).

The authors devote a substantial effort into characterizing immune responses elicited by vaccination with Δ menT4 Δ T3 in mice and trying to link these immune features as correlates of protection. However, data presented in Fig.4 demonstrate that in mice, protection was either worse (Fig. 4B), similar, or moderately improved (Fig. 4E) with the Δ menT4 Δ T3 vaccine. Though fewer reagents for cell characterization of guinea pigs are available, it would have been more appropriate to characterize responses in the animal model where enhanced protection was observed. Any data taken from the mice may be difficult to extrapolate to the guinea pig.

*We thank the reviewer for his/her in-depth review and valuable suggestions for improving the quality of the manuscript. As rightly pointed out by the reviewer, the lack of reagents makes it challenging to study immunological profiles in guinea pigs. Therefore, the mice model of infection is preferred over guinea pigs to study immunological responses. The immune correlates of protection of a vaccine candidate in mice have been used as standards for a number of infectious diseases including *M. tuberculosis*. As suggested by the editor and reviewer, we have included a lack of characterization of immune response studies in guinea pigs as a limitation of the study.*

Minor Comments

In the Materials and Methods section, the authors describe flow cytometry and ELISA experiments yet do not state which animal species these experiments are being performed on. This should be clarified.

As suggested, we have clarified this in the revised manuscript.

On page 13, the final sentence of the first paragraph, there may be a typo. The sentence states "...reduced levels of transcripts encoding for proteins required for either in vitro growth of stress...". The authors did not demonstrate that the Δ menT4 Δ T3 mutations impact growth in vitro, so perhaps they meant in vivo?

As suggested, we have corrected this in the revised manuscript.

The authors make several attempts to complement the Δ menT4 Δ T3 double mutation, yet in some circumstances cannot. While this could be due to circumstances proposed by the authors, the authors cannot rule out that this is also indicative of a second site mutation in the mutant and should discuss this.

*In the present study, we have generated Δ menT3, Δ menT4 and Δ menT4 Δ 3 strains of *M. tuberculosis*. The deletion of menT3 and menT4 in the genome of *M. tuberculosis* was confirmed by both PCR whole-genome sequencing. As shown in Fig. S2, the sequencing reads were mapped to both menT3 and menT4 neighbouring genes. Several studies have shown that the replacement*

of an open reading frame with an antibiotic selection marker might result in a polar effect on the expression of neighbouring genes. To rule out this possibility, we have now performed qPCR studies and report that the relative levels of *menT3* and *menT4* neighbouring genes were comparable in the wild type and $\Delta menT4\Delta T3$ double mutant strain. Also, we have included a table with a complete list of detected genes and their respective *p*-values and *q*-values (Table S3). Another possibility for the observed attenuated phenotype could be the loss of virulence-associated lipids such as PDIMs. As shown in the original manuscript, we observed that the lipid profiles of wild type and $\Delta menT4\Delta T3$ were comparable. As suggested by the reviewer we have also included data concerning the characterization of single mutant strains. We show that the growth of wild type, $\Delta menT3$ and $\Delta menT4$ strains of *M. tuberculosis* was comparable in stress conditions and in THP-1 macrophages. Previously, transposon site hybridization (TRASH) and Designer array for defined mutant analysis (DeADMAN) have been performed to identify genes that are required for *in vivo* growth of *M. tuberculosis*. In these studies, *menT3* and *menT4* were not identified to be required for *M. tuberculosis* to establish infection in mice (Sasseti et al., 2003; Lamichanne et al., 2005). Based on these observations, we conclude that both *MenT3* and *MenT4* are mutually redundant and a phenotype is observed upon deletion of both *menT3* and *menT4* in *M. tuberculosis*. In the present study, we show that simultaneous deletion of both *menT3* and *menT4* resulted in increased susceptibility of *M. tuberculosis* upon exposure to oxidative stress. We also show that compared to the parental strain, $\Delta menT4\Delta T3$ strain was attenuated for growth in mice and guinea pigs. We have included this information in the revised manuscript.

Vaccination and challenge experiments presented in Figure 4 represent data from mice and guinea pigs, yet when these experiments are described in the text (starting on the top of page 16), it is unclear to the reader that the first set of experiments was performed in mice (and in which mouse strain). This should be clarified in the text.

As suggested, we have clarified this in the revised manuscript.

On page 21, the authors conclude that because the $\Delta menT4\Delta T3$ mutant did not have a growth defect in macrophages, that macrophages are unable to mimic the conditions encountered by *M. tuberculosis* inside host tissues. This is a strained interpretation given that there are many examples of mutants of *M. tuberculosis* that are attenuated for growth in macrophages and that growth defect is directly linked to attenuation *in vivo*. A better interpretation may be that the functions of *MenT4* and *MenT3* lie outside of growth in macrophages, which still does not rule out the importance of these factors for interactions with many other host cells and aspects of the host environment.

*As rightly pointed by the reviewer, in addition to macrophages, it has been reported that *M. tuberculosis* is able to infect other cell types such as lung epithelial cells, dendritic cells, adipocytes, neutrophils, and mesenchymal stem cells (Lyadova, 2017, Scordo et al., 2016, Khan et al., 2017, Beigier et al., 2017, Mihret, 2012). Since, the growth kinetic for wild type and $\Delta menT4\Delta T3$ is comparable in THP-1 macrophages, there is a possibility that the function of *MenT3* and *MenT4* lies outside macrophages and is involved in the interaction of *M. tuberculosis* with other cells involved in host-pathogen interactions. We have included this information in the revised manuscript.*

In the Discussion, that authors propose a model for the proposed biological mechanism for the in vivo phenotypes of the Δ menT4 Δ T3 mutant. While these are reasonable hypotheses, putting them into such a figure imparts to the reader that many aspects of this model have been directly addressed in these studies with the Δ menT4 Δ T3 mutant, such as the role of altered calcium signaling, changes in the Th1 responses etc, and they have not. I might suggest not including this figure.

We thank the reviewer for this comment. We are of the opinion that this figure summarizes the major findings of our manuscript. Therefore, we have retained this figure in the revised manuscript. However, if the reviewer insists, we can remove this figure from the revised manuscript.

REVIEWER COMMENTS

Reviewer #1 (Remarks to the Author):

The authors have responded to my all comments, clarifying the most important points in their reviewed manuscript.

Reviewer #3 (Remarks to the Author):

The genome of *M. tuberculosis* encodes more than 90 toxin-antitoxin genes, but whether these genes are all transcribed into functional products, and what roles these products play in the general biology and pathogenesis of *M. tuberculosis* is yet unknown. The authors attempt to address this gap by examining the role of toxin-antitoxin MenT3 and MenT4 by creating menT3 and menT4 deficient *M. tuberculosis*. Overall, the authors demonstrate that MenT3/T4 play a role in combating oxidative stress, and in promoting growth in vivo. Further, the authors demonstrate that these attenuated *M. tuberculosis* Δ menT4 Δ T3 mutant can be employed as a vaccine that can provide enhanced protection compared to the gold standard BCG in guinea pigs in terms of reduction of bacterial burdens in organs and reduction in lung inflammation. Benefits of the Δ menT4 Δ T3 vaccine were less readily observed in mice, though the authors devote much effort into characterizing the cellular immune responses in mice to the Δ menT4 Δ T3 vaccine construct. The authors also draw conclusions about the mechanism of enhanced protection based on their findings, though these conclusions are not supported by direct follow-up studies. Overall, the authors provide important new information about the role of this toxin-antitoxin pair in the in vitro and in vivo lifestyles of *M. tuberculosis* and provide encouraging data that the Δ menT4 Δ T3 mutant could be useful as a novel vaccine candidate.

The authors submitted a revised manuscript that addresses some, but not all, or the major critiques of this study, and those remaining issues are described below.

Major Comments

1. The authors were asked to provide missing information about animal studies, and much of this information has still not been provided. Reproducibility of animal studies hinges on many variables, such as age and sex of the animals, the origin of the animals (vendor), housing conditions and food conditions etc. None of these details have been provided. The authors provided information about the number of replicate experiments in the rebuttal, but not in the manuscript itself. This information should be included in the manuscript, especially to highlight that vaccine efficacy experiments were only performed once, a fact that also draws concern about the reproducibility of these findings and extrapolation to the large pool of immunological data in the manuscript.
2. The authors provided some discussion about their choice of using C57BL/6 animals but did not address why they chose Balb/c mice, a strain with different inherent susceptibility and a more Th2 skewed immunophenotype, to perform their initial challenge studies. Variable incorporation of mouse strains makes comparisons between primary challenge and protection experiments more difficult.
3. While I agree that immunological studies in guinea pigs are difficult due to the lack of reagents, the underlying issue still remains that the authors go to great lengths to characterize immune correlates in

the mouse system and to draw conclusions about correlates of protection based on studies where protection in mice was either worse (Fig. 4B), similar, or moderately improved (Fig. 4E) with the $\Delta\text{menT4}\Delta\text{T3}$ vaccine. Also, given that this is a one-off experiment, it is also difficult to assess the reproducibility of these findings. If the vaccine is not more protective than BCG, then how can one draw conclusions about whether any of the immunological findings are important for protection? Beyond what is already known about the importance of antigen-specific T cells.

Minor Comments

4. The authors provided some rationale and further details about their attempts to complement the double mutation and assessments of this strain. None of the data or information that the authors provided can address the possibility that a second site suppressor mutation at a distal location could have occurred during the creation of the double mutant. Second site suppressor mutations are often in genes that are not proximal to the targets of the mutation, and therefore analyses of the adjacent genes would not uncover such mutations. This caveat should be added to the manuscript. The authors also don't mention whether they confirmed expression of MenT3 and MenT4 in their complemented strain. The authors also mention methodologies derived from Dr. Jacobs to generate knockouts via mycobacteriophages. Then the authors should also be familiar with technology developed by that same laboratory to generate in-frame, unmarked mutations by double allelic exchange. That methodology is more favorable for complementation because the resulting mutations are unmarked, and therefore all antibiotics are available for use in plasmids for complementation. Should the authors continue to be interested in this strain, I would suggest recreating the strain with unmarked deletions and performing complementation in that strain.

Response to the queries raised by reviewers

We thank the reviewers for their careful assessment of our manuscript. We found their comments and suggestions useful, which enabled us to further improve the manuscript. In the revised manuscript, we have addressed the comments raised by reviewer 3. In the responses below, the reviewers' comments are in bold, and our responses are indented in italics. We sincerely hope that the revised manuscript will now be considered suitable for publication in Nature Communications.

Reviewer #1 (Remarks to the Author):

The authors have responded to my all comments, clarifying the most important points in their reviewed manuscript.

We sincerely thank the reviewer for accepting our clarifications in the revised manuscript.

Reviewer #3 (Remarks to the Author):

The genome of *M. tuberculosis* encodes more than 90 toxin-antitoxin genes, but whether these genes are all transcribed into functional products, and what roles these products play in the general biology and pathogenesis of *M. tuberculosis* is yet unknown. The authors attempt to address this gap by examining the role of toxin-antitoxin MenT3 and MenT4 by creating *menT3* and *menT4* deficient *M. tuberculosis*. Overall, the authors demonstrate that MenT3/T4 play a role in combating oxidative stress, and in promoting growth in vivo. Further, the authors demonstrate that these attenuated *M. tuberculosis* Δ *menT4* Δ *T3* mutant can be employed as a vaccine that can provide enhanced protection compared to the gold standard BCG in guinea pigs in terms of reduction of bacterial burdens in organs and reduction in lung inflammation. Benefits of the Δ *menT4* Δ *T3* vaccine were less readily observed in mice, though the authors devote much effort into characterizing the cellular immune responses in mice to the Δ *menT4* Δ *T3* vaccine construct. The authors also draw conclusions about the mechanism of enhanced protection based on their findings, though these conclusions are not supported by direct follow-up studies. Overall, the authors provide important new information about the role of this toxin-antitoxin pair in the in vitro and in vivo lifestyles of *M. tuberculosis* and provide encouraging data that the Δ *menT4* Δ *T3* mutant could be useful as a novel vaccine candidate. The authors submitted a revised manuscript that addresses some, but not all, or the major critiques of this study, and those remaining issues are described below.

We thank the reviewer for carefully assessing the manuscript. We also apologise to the reviewer for not addressing all comments in the previous revised manuscript. In the revised manuscript, we have addressed the concerns raised by the reviewer and hope that the manuscript will now be considered suitable for publication in Nature Communications.

Major Comments

1. The authors were asked to provide missing information about animal studies, and much of this information has still not been provided. Reproducibility of animal studies hinges on many variables, such as age and sex of the animals, the origin of the animals (vendor),

housing conditions and food conditions etc. None of these details have been provided. The authors provided information about the number of replicate experiments in the rebuttal, but not in the manuscript itself. This information should be included in the manuscript, especially to highlight that vaccine efficacy experiments were only performed once, a fact that also draws concern about the reproducibility of these findings and extrapolation to the large pool of immunological data in the manuscript.

We sincerely apologise to the reviewer for not providing this information in the earlier version of the manuscript. As suggested, we have now included this information in the revised manuscript. In the revised manuscript, we have added the following in Animal husbandry details in the Materials and Methods section:

6-8 weeks old outbred female Duncan Hartley guinea pigs (~250-300g) were obtained from Disease Free Small Animal House, Lala Lajpat Rai University of Veterinary and Animal Sciences, Hisar. 6-8 weeks old inbred female Balb/c and C57BL/6 mice (~ 20-25g) were obtained from the Experimental Animal Facility, NCR Biotech Science Cluster, Faridabad. Animals were housed in a group of either 5 (mice) or 3 (guinea pigs) in individually ventilated cages in BSL-3 labs. The animals were maintained at a room temperature of $22 \pm 3^{\circ}\text{C}$, relative humidity of 30 to 70%, 15-20 air changes/hours, light intensity of 325-350 lux with a 14 h light/10 h dark cycle and noise intensity of <85 db. A sterilized pellet diet (Altromin International, Germany) was offered ad libitum to the animals throughout the experimental period. Ad libitum aqua guard filtered and autoclaved water was served in bottles fitted with stainless steel nozzles.

As suggested, we have also provided information about the number of replicate experiments in the revised manuscript. We again sincerely apologise to the reviewer for this oversight and for not providing the required information in the earlier version of the manuscript.

2. The authors provided some discussion about their choice of using C57BL/6 animals but did not address why they chose Balb/c mice, a strain with different inherent susceptibility and a more Th2 skewed immunophenotype, to perform their initial challenge studies. Variable incorporation of mouse strains makes comparisons between primary challenge and protection experiments more difficult.

We acknowledge that Balb/c and C57BL/6 possess a T_{H2} and T_{H1} skewed immunophenotype, respectively. The selection of Balb/c mice for pathogenesis studies is a routine practice in the laboratory, as also evident from our other recently published studies (Chugh et al., 2024, Chauhan et al., 2023, Tiwari et al., 2022; Agarwal et al., 2020). Given that we observed an attenuated phenotype for $\Delta\text{menT4}\Delta\text{T3}$ relative to the parental strain in Balb/c mice and were further evaluating this strain as a vaccine candidate (as discussed in the manuscript), detailed immunological studies were performed in C57BL/6 mice, a preferred mouse model for immune studies. We had also determined the bacillary loads of the immunizing bacilli at six weeks post-immunization before challenging the animals with H37Rv. We observed that the number of immunizing bacilli (BCG and $\Delta\text{menT4}\Delta\text{T3}$) in the lungs and spleens of C57BL/6 mice and guinea pigs were below the detection limit. We have included this information in the revised manuscript.

3. While I agree that immunological studies in guinea pigs are difficult due to the lack of reagents, the underlying issue still remains that the authors go to great lengths to characterize immune correlates in the mouse system and to draw conclusions about

correlates of protection based on studies where protection in mice was either worse (Fig. 4B), similar, or moderately improved (Fig. 4E) with the Δ menT4 Δ T3 vaccine. Also, given that this is a one-off experiment, it is also difficult to assess the reproducibility of these findings. If the vaccine is not more protective than BCG, then how can one draw conclusions about whether any of the immunological findings are important for protection? Beyond what is already known about the importance of antigen-specific T cells.

*In the present study, we show that immunization of guinea pigs with Δ menT4 Δ T3 imparted significantly better protection compared to immunization with BCG. We also show that immunization of C57BL/6 mice with Δ menT4 Δ T3 was also able to impart protection against challenge with *M. tuberculosis*. As rightly pointed out by the reviewer, these levels were either worse or similar or moderately improved relative to immunization with BCG. The levels of protection observed upon immunization with Δ menT4 Δ T3 were similar to those reported for other attenuated vaccine candidates such as Δ leuD (in guinea pigs) or Δ panCD (in guinea pigs) or Δ RD1 Δ panCD (in C57BL/6 mice). In another study, T_H1 type immune response has been reported in BCG-vaccinated guinea pigs (Tree et al., 2006). It has also been shown that C57BL/6 mice are suitable for studying T_H1 cell-based pathology, and this pathology is driven by the generation of IFN- γ response. Previous studies have shown that protective immunity against TB depends on an acquired cellular immune response involving T-cell subsets, and a T_H1 -type response is considered favourable in imparting protection against *M. tuberculosis* (Barnes and Venkayalapati, 2005). Given that immunization with Δ menT4 Δ T3 imparted protection in mice and guinea pigs and there is lack of reagents to study immune responses in guinea pigs, we performed detailed immunological studies in C57BL/6 mice. In our immune response studies, we observed increased activated memory T cell response in Δ menT4 Δ T3 immunized mice as compared to the BCG immunized or naive mice. Further, we also observed a significant increase in the secretion of T_H1 cytokines such as IFN- γ , TNF- α and IL-2 in culture supernatants of PPD-stimulated splenocytes from Δ menT4 Δ T3 immunized mice, in agreement with the flow cytometry data. These observations are in agreement with previous studies that T_H1 immune responses are necessary for host defence against TB. As suggested, we have also included the number of replicates for each animal experiment in the revised manuscript.*

Minor Comments

4. The authors provided some rationale and further details about their attempts to complement the double mutation and assessments of this strain. None of the data or information that the authors provided can address the possibility that a second site suppressor mutation at a distal location could have occurred during the creation of the double mutant. Second site suppressor mutations are often in genes that are not proximal to the targets of the mutation, and therefore analyses of the adjacent genes would not uncover such mutations. This caveat should be added to the manuscript. The authors also don't mention whether they confirmed expression of MenT3 and MenT4 in their complemented strain. The authors also mention methodologies derived from Dr. Jacobs to generate knockouts via mycobacteriophages. Then the authors should also be familiar with technology developed by that same laboratory to generate in-frame, unmarked mutations by double allelic exchange. That methodology is more favorable for complementation because the resulting mutations are unmarked, and therefore all antibiotics are available for use in

plasmids for complementation. Should the authors continue to be interested in this strain, I would suggest recreating the strain with unmarked deletions and performing complementation in that strain.

*We thank the reviewer for this comment. We have now included qPCR data showing that the relative transcript levels of toxins, *menT3* and *menT4*, were restored in their respective complemented strains. Also, we have added the statement pertaining to secondary site mutation in the $\Delta menT4\Delta T3$ strain in the discussion section of the revised manuscript. As rightly pointed out by the reviewer, future experiments would include unmarking and construction of $\Delta menT4\Delta T3$ -based multiple-allele mutant strains (such as *panCD*, *leuD*, *metX*, etc.), and these strains would be evaluated for safety and efficacy studies in compliance with the Geneva consensus. We have included these statements in the revised manuscript for better clarity.*